# GI-GS: Global Illumination decomposition on Gaussian Splatting for Inverse Rendering

**Hongze Chen, Zehong Lin**[*]**, Jun Zhang**
The Hong Kong University of Science and Technology
hchenec@connect.ust.hk, eezhlin@ust.hk, eejzhang@ust.hk

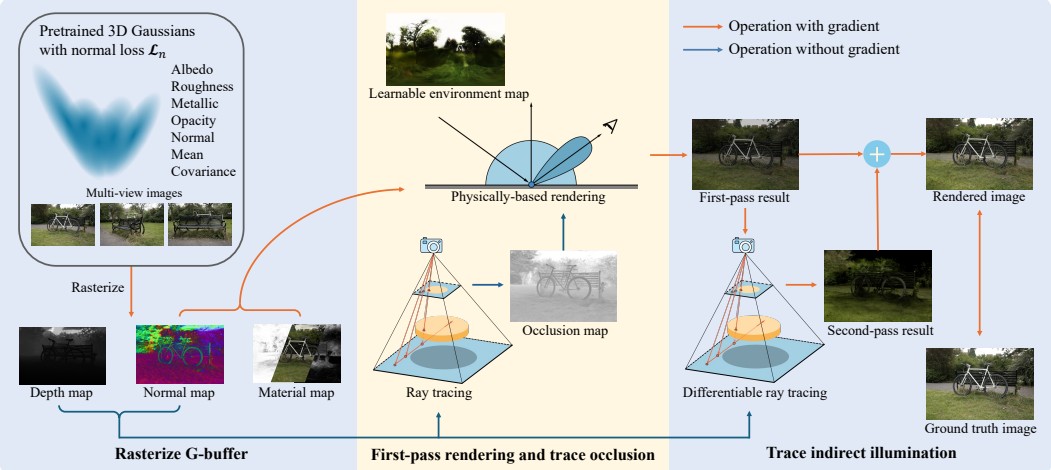

Figure 1: **Overview of GI-GS.** GI-GS takes input a set of pretrianed 3D Gaussians, each with a normal attribute. It first rasterizes the scene geometry and materials into a G-buffer. Next, it incorporates a differentiable PBR pipeline to obtain the rendering result under *direct* lighting and performs path tracing to model the occlusion. Finally, it employs differentiable ray tracing to calculate *indirect* lighting from the scene geometry and the previous rendering result. The final rendered image is a fusion of the first-pass and second-pass results and uses the ground truth image for supervision.

## ABSTRACT

We present *GI-GS*, a novel inverse rendering framework that leverages 3D Gaussian Splatting (3DGS) and deferred shading to achieve photo-realistic novel view synthesis and relighting. In inverse rendering, accurately modeling the shading processes of objects is essential for achieving high-fidelity results. Therefore, it is critical to incorporate global illumination to account for indirect lighting that reaches an object after multiple bounces across the scene. Previous 3DGS-based methods have attempted to model indirect lighting by characterizing indirect illumination as learnable lighting volumes or additional attributes of each Gaussian, while using baked occlusion to represent shadow effects. These methods, however, fail to accurately model the complex physical interactions between light and objects, making it impossible to construct realistic indirect illumination during relighting. To address this limitation, we propose to calculate indirect lighting using efficient path tracing with deferred shading. In our framework, we first render a G-buffer to capture the detailed geometry and material properties of the scene. Then, we perform physically-based rendering (PBR) only for direct lighting. With the G-buffer and previous rendering results, the indirect lighting can be calculated through a lightweight path tracing. Our method effectively models indirect lighting under any given lighting conditions, thereby achieving better novel view synthesis and competitive relighting. Quantitative and qualitative results show that our GI-GS outperforms existing baselines in both rendering quality and efficiency. Project page: https://stopaimme.github.io/GI-GS-site/.

---

[*]Corresponding author

# 1 INTRODUCTION

Inverse rendering (Barrow et al., 1978) aims to estimate the physical attributes of a 3D scene, such as material properties, geometry, and lighting, from captured images. This task is crucial for various downstream applications, including novel view synthesis, relighting, and VR/AR. Similar to many inverse problems, inverse rendering is often considered ill-posed due to unknown lighting conditions and the intractability of the integrals in the rendering equation. Recent research has adopted the Neural Radiance Fields (NeRF) (Mildenhall et al., 2021) paradigm, which represents scenes as neural fields using multi-layer perceptrons (MLPs). Nonetheless, rendering an image with NeRF-like methods requires dense querying of MLPs along ray directions, which significantly affects the rendering speed. Moreover, the limited representational capacity of MLPs hinders their ability to faithfully model physical attributes of scenes and complex interactions between light and surfaces.

More recently, 3D Gaussian Splatting (3DGS) (Kerbl et al., 2023) has gained significant attention for its superior reconstruction quality and rendering speed. This method explicitly represents scenes using a set of 3D Gaussians and employs a tile-based rasterization pipeline for efficient rendering. Unlike NeRF, which utilizes MLPs to encode view-dependent colors across the scene, 3DGS assigns spherical harmonics to each Gaussian point to capture these view-dependent effects. Nevertheless, 3D Gaussians trained under specific lighting conditions cannot be used to render relighted results under different lighting setups, as the view-dependent colors entangle the objects' materials with the training lighting conditions. This constraint restricts the broader applicability of 3DGS. To address this limitation, recent studies (Jiang et al., 2024; Shi et al., 2023; Liang et al., 2024; Gao et al., 2023; Wu et al., 2024) have proposed several inverse rendering frameworks based on 3DGS. These methods successfully decouple color into the interactions between lighting and surface materials by incorporating physically-based rendering (PBR) techniques. Some of them have also attempted to model indirect lighting, but they typically store the indirect lighting either statically in a 3D Gaussian or in additional volumes and optimize it during training. These approaches overlook the dynamic nature of light and, therefore, cannot accurately model indirect lighting during the relighting process.

To overcome these limitations, we propose *GI-GS*, a novel inverse rendering framework based on 3DGS that achieves accurate estimation of materials and geometry, as well as global illumination decomposition. Our key innovation is to calculate indirect lighting by efficient path tracing based on the deferred shading technique. Specifically, as shown in Fig. 1, we first rasterize the geometry and materials of the scene, storing the results in a G-buffer. Then, we incorporate a PBR pipeline to efficiently compute the complex rendering equations and thoroughly decompose the illumination. We employ a learnable environment map to approximate direct lighting. For indirect lighting and occlusion, instead of modeling them as attributes of 3D Gaussian points or using baked volumes as in previous methods, we calculate them from the first-pass rendering results through path tracing. This approach ensures that our indirect lighting is more consistent with the rendering equation. Consequently, our approach can reconstruct high-fidelity geometry and materials of a complex real scene under unknown natural illumination, thereby achieving state-of-the-art rendering of novel view synthesis and enabling additional applications like relighting. Notably, to the best of our knowledge, we are among the first to develop a 3DGS-based inverse rendering framework capable of modeling indirect lighting during the relighting process.

In summary, our main contributions are as follows:

- We present GI-GS, a novel inverse rendering framework based on 3DGS that achieves state-of-the-art intrinsic decomposition and rendering results for both objects and scenes.
- We propose an efficient tile-based path tracing method based on deferred shading, which enables accurate occlusion and indirect lighting calculation from previously rendered results stored in the G-buffer.
- We extend our path tracing-based indirect lighting from screen space to world space for scene-level datasets, achieving more natural occlusion and indirect lighting.

# 2 RELATED WORKS

**Neural Rendering and Radiance Fields.** Neural rendering, exemplified by NeRF (Mildenhall et al., 2021), has received widespread attention in recent years due to its remarkable representational

capabilities and support for end-to-end optimization. NeRF implicitly represents a scene as a continuous radiance field using MLPs that take the 3D positions and view directions as inputs to output the density and view-dependent color. However, the rendering process of NeRF requires dense importance sampling along rays, which involves repeated MLP queries and consequently limits rendering speed. To address this issue, subsequent research has introduced efficient explicit data structures, such as hash grids (Müller et al., 2022), voxels (Sun et al., 2022; Fridovich-Keil et al., 2022), and tri-planes (Chen et al., 2022), to encode features in 3D space, thereby accelerating both training and inference phases. In addition, some works (Barron et al., 2021; 2022; 2023) have focused on enhancing the rendering quality of NeRF, while others (Wang et al., 2021; Yariv et al., 2021) have utilized radiance fields to represent signed distance functions (SDFs) for surface reconstruction.

Recently, 3DGS (Kerbl et al., 2023) replaces MLPs with a set of 3D Gaussian primitives to explicitly represent the neural field and incorporates a tile-based rasterization pipeline to achieve high-quality real-time rendering. Since 3DGS is originally designed for novel view synthesis and lacks relevant geometric constraints or priors, how to reconstruct accurate geometry from 3DGS remains an important problem. Subsequent methods have significantly improved the quality of the reconstructed geometry by integrating geometrically relevant regularization terms (Guédon & Lepetit, 2024) or employing more geometrically accurate representations (Huang et al., 2024; Dai et al., 2024). In our work, we extend 3DGS to the inverse rendering task by combining it with PBR, thereby broadening its applicability to a wider range of downstream tasks.

**Inverse Rendering.** Recovering geometry, materials, and lighting from captured images has been a long-standing challenge due to the unknown lighting conditions and the complex interactions between light and objects. Most previous inverse rendering methods simplify the problem by introducing prior information. For instance, Barron & Malik (2014); Li et al. (2018) introduce plane assumptions to simplify the acquisition of normals, Bi et al. (2020b); Nam et al. (2018) use a flashlight to align the light source with the camera positions, and Xia et al. (2016) capture a rotating object under fixed camera and lighting conditions.

Inspired by the success of volume rendering techniques based on radiance fields, subsequent works (Bi et al., 2020a; Srinivasan et al., 2021; Boss et al., 2021; Zhang et al., 2021b; Jin et al., 2023; Zhang et al., 2021a; 2022; Yao et al., 2022) have attempted to apply the NeRF paradigm to address this issue by leveraging the powerful representational capabilities of neural networks. For instance, Neural Reflectance Fields (Bi et al., 2020a) is the first work to introduce the bidirectional reflectance distribution function (BRDF) and lighting in NeRF, which represents a scene as a reflectance field under a point light source. NeRV (Srinivasan et al., 2021) uses an environment map to represent the given light condition and trains an MLP to approximate the visibility of a given point in space from different directions, while PhySG (Zhang et al., 2021a) models lighting as spherical Gaussians and incorporates SDFs for more accurate geometry reconstruction. Besides, NeRFactor (Zhang et al., 2021b) adopts a multi-stage training to reduce the difficulty of optimization. TensoIR (Jin et al., 2023) employs a compact tri-plane representation for ray tracing-based indirect lighting calculations. Moreover, Neilf (Yao et al., 2022) utilizes a neural incident light field to better model the light source.

Recently, some works (Jiang et al., 2024; Wu et al., 2024) have extended 3DGS to the inverse rendering task by binding BRDFs as attributes of Gaussians. Besides, GS-IR (Liang et al., 2024) leverages baking-based volumes to store occlusion and indirect illumination. Relightable 3D Gaussian (Gao et al., 2023) bakes the occlusion by using point-based ray tracing and parameterizes the indirect lighting as additional spherical harmonics for each Gaussian. Nevertheless, both methods rely on baking to accelerate rendering and cannot model indirect lighting during relighting. In contrast, we propose a 3DGS-based framework for inverse rendering and leverage deferred shading to realize efficient path tracing-based occlusion and indirect illumination for both rendering and relighting.

## 3 PRELIMINARIES

**3D Gaussian Splatting.** 3DGS (Kerbl et al., 2023) explicitly represents a scene as a set of 3D Gaussian primitives. Each primitive is parameterized by a Gaussian distribution as:

$$G(\boldsymbol{x}) = e^{-\frac{1}{2}(\boldsymbol{x}-\boldsymbol{\mu})^T \boldsymbol{\Sigma}^{-1}(\boldsymbol{x}-\boldsymbol{\mu})}, \tag{1}$$

where $\boldsymbol{\mu} \in \mathbb{R}^3$ denotes the mean vector and $\boldsymbol{\Sigma} \in \mathbb{R}^{3\times3}$ denotes the covariance matrix. Besides, each 3D Gaussian primitive is assigned a set of spherical harmonics to model the view-dependent color

and opacity $\alpha$ for volume rendering. During the rendering process, the 3D Gaussians are projected onto the 2D image plane through EWA splatting (Zwicker et al., 2001). Specifically, given the view transformation matrix $\boldsymbol{W}$ and the Jacobian of the affine approximation of the projective transformation $\boldsymbol{J}$, the covariance matrix $\boldsymbol{\Sigma}'$ in camera coordinates is computed as $\boldsymbol{\Sigma}' = \boldsymbol{JW\Sigma W^T J^T}$. For each screen pixel, its color is obtained by $\alpha$-blending 2D Gaussians sorted by depth as:

$$C = \sum_{i \in \mathcal{N}} T_i c_i \alpha_i \text{ with } T_i = \prod_{j=1}^{i-1} (1 - \alpha_j), \tag{2}$$

where $\mathcal{N}$ denotes the set of Gaussians, $\alpha_i$ is the transmittance of each Gaussian derived from its opacity and covariance matrix, and $T_i$ is the accumulated transmittance.

**Physically-based Rendering.** PBR is a rendering approach that models the color of an object as the interaction of light and surface material. Given a surface point $\boldsymbol{x}$ and its normal vector $\boldsymbol{n}$, the outgoing light $L_o$ in the camera view direction $\boldsymbol{\omega}_o$ can be calculated using the following rendering equation (Kajiya, 1986):

$$L_o(\boldsymbol{\omega}_o, \boldsymbol{x}) = \int_\Omega f_r(\boldsymbol{\omega}_o, \boldsymbol{\omega}_i, \boldsymbol{x}) L_i(\boldsymbol{\omega}_i, \boldsymbol{x})(\boldsymbol{\omega}_i \cdot \boldsymbol{n}) d\boldsymbol{\omega}_i, \tag{3}$$

where $\Omega$ is the upper hemisphere centered at $\boldsymbol{x}$ and $f_r(\boldsymbol{\omega}_o, \boldsymbol{\omega}_i, \boldsymbol{x})$ is the bidirectional reflectance distribution function (BRDF), which characterizes the relationship between the outgoing light $L_o$ and the incident light $L_i$ coming from direction $\boldsymbol{\omega}_i$ based on the material of the object. The BRDF can be divided into a diffuse component $f_d$ and a specular component $f_s$ according to the Cook-Torrance BRDF model (Cook & Torrance, 1982) as follows:

$$f_r(\boldsymbol{\omega}_i, \boldsymbol{\omega}_o) = \underbrace{(1-m)\frac{\boldsymbol{a}}{\pi}}_{\text{diffuse } f_d} + \underbrace{\frac{D(\boldsymbol{h};\rho)F(\boldsymbol{\omega}_o,\boldsymbol{h};\boldsymbol{a},m)G(\boldsymbol{\omega}_i,\boldsymbol{\omega}_o,\boldsymbol{h};\rho)}{4(\boldsymbol{n}\cdot\boldsymbol{\omega}_i)(\boldsymbol{n}\cdot\boldsymbol{\omega}_o)}}_{\text{specular } f_s(\boldsymbol{\omega}_i,\boldsymbol{\omega}_o)}, \tag{4}$$

where $\boldsymbol{a} \in [0,1]^3$ denotes the diffuse albedo, $m \in [0,1]$ denotes the metallic value, $\rho \in [0,1]$ denotes the roughness, and $\boldsymbol{h} = \frac{\boldsymbol{\omega}_i + \boldsymbol{\omega}_o}{\|\boldsymbol{\omega}_i + \boldsymbol{\omega}_o\|}$ is the half vector. The specular component $f_s(\boldsymbol{\omega}_i, \boldsymbol{\omega}_o)$ is further decomposed into a normal distribution function (NDF) $D$, a Fresnel term $F$, and a geometry term $G$. Consequently, the outgoing irradiance $L_o$ can be decomposed into diffuse light $L_d$ and specular light $L_s$ as follows:

$$L_o(\boldsymbol{\omega}_o, \boldsymbol{x}) = L_d(\boldsymbol{x}) + L_s(\boldsymbol{\omega}_o, \boldsymbol{x}), \tag{5}$$

where

$$L_d(\boldsymbol{x}) = \int_\Omega f_d L_i(\boldsymbol{\omega}_i, \boldsymbol{x})(\boldsymbol{\omega}_i \cdot \boldsymbol{n}) d\boldsymbol{\omega}_i, \; L_s(\boldsymbol{\omega}_o, \boldsymbol{x}) = \int_\Omega f_s(\boldsymbol{\omega}_i, \boldsymbol{\omega}_o) L_i(\boldsymbol{\omega}_i, \boldsymbol{x})(\boldsymbol{\omega}_i \cdot \boldsymbol{n}) d\boldsymbol{\omega}_i. \tag{6}$$

**Global Illumination Decomposition.** In real-world scenes, the incident light reaching a surface is often a combination of direct illumination from light sources and indirect illumination from light reflected off other objects. This is due to the occlusion of the light sources by the environment or the object itself, as well as multiple bounces of light between objects. To achieve a more realistic rendering of these scenes, for the diffuse component $L_d$, we decompose the incident light into direct illumination and indirect illumination, and calculate them separately as follows:

$$\begin{aligned} L_d(\boldsymbol{x}) &= \int_\Omega (1-m)\frac{\boldsymbol{a}}{\pi} L_i(\boldsymbol{\omega}_i, \boldsymbol{x})(\boldsymbol{\omega}_i \cdot \boldsymbol{n}) d\boldsymbol{\omega}_i \\ &= (1-m)\frac{\boldsymbol{a}}{\pi}\left(\int_{\Omega_{\text{Vis}}} L_i^{dir}(\boldsymbol{\omega}_i, \boldsymbol{x})(\boldsymbol{\omega}_i \cdot \boldsymbol{n}) d\boldsymbol{\omega}_i + \int_{\Omega_{\text{Occ}}} L_i^{ind}(\boldsymbol{\omega}_i, \boldsymbol{x})(\boldsymbol{\omega}_i \cdot \boldsymbol{n}) d\boldsymbol{\omega}_i\right) \\ &\approx (1-m)\frac{\boldsymbol{a}}{\pi}O(\boldsymbol{x})I_{dir}(\boldsymbol{x}) + (1-m)\frac{\boldsymbol{a}}{\pi}I_{ind}(\boldsymbol{x}), \end{aligned} \tag{7}$$

where the two integrals calculate the irradiance of the incident direct lighting and indirect lighting in the visible area $\Omega_{\text{Vis}}$ and the occluded area $\Omega_{\text{Occ}}$ in the upper hemisphere, respectively. Here, we have $\Omega_{\text{Vis}} \cup \Omega_{\text{Occ}} = \Omega$ and $\Omega_{\text{Vis}} \cap \Omega_{\text{Occ}} = \emptyset$. Furthermore, we can approximate these two integrals using the direct irradiance $I_{dir}$ and the indirect irradiance $I_{ind}$, and introduce an occlusion term $O(\boldsymbol{x})$ to approximate the visibility.

# 4 METHOD

In this section, we present our *GI-GS* framework for inverse rendering, as depicted in Fig. 1. Given a set of posed RGB images, the pipeline proceeds in three stages. In the first stage, we follow the vanilla 3DGS pipeline to reconstruct the scene geometry. We incorporate the normal as a new attribute for each Gaussian primitive and optimize it using pseudo normals derived from the depth map (see Sec. 4.1). In the second stage, to recover the BRDF and lighting conditions, we incorporate a PBR pipeline with the deferred shading technique. We first render the depth map, normal map, and BRDF maps from the 3D Gaussians and store them in a G-buffer. Then, we estimate the occlusion from the depth map and normal map, and leverage a differentiable PBR pipeline to render an image $I_{dir}$ under direct lighting using a learnable environment map (see Sec. 4.2). In the final stage, given the geometry (depth and normals) and the first-pass rendered image $I_{dir}$, we apply differentiable path tracing to obtain the rendering result $I_{ind}$ under indirect lighting (see Sec. 4.3). Finally, we combine the two rendering results and optimize the lighting and the Gaussians' BRDF attributes.

## 4.1 GEOMETRY RECONSTRUCTION

To enable PBR, we first need to reconstruct reliable scene geometry from 3DGS.

**Depth Rendering.** We follow GS-IR (Liang et al., 2024) to treat the depth as a linear interpolation of the depth of the $N$ Gaussians, i.e., $d = \sum_{i=1}^{N} w_i d_i$, where $w_i = \frac{T_i \alpha_i}{\sum_{i=1}^{N} T_i \alpha_i}$. Compared to directly $\alpha$-blending the depth of each Gaussian, this approach reduces artifacts and ensures a smoother transition in rendered depth values between the deepest and shallowest Gaussians.

**Normal Estimation.** In this work, we treat the normal $\boldsymbol{n}$ as an attribute of each Gaussian and use $\alpha$-blending to render the normal map. Unlike existing methods (Shi et al., 2023; Chen et al., 2024) that use the direction of the shortest axis of the ellipsoid as the normal direction and apply regularization to force the Gaussian into a flat disk, our approach avoids the degradation in rendering quality caused by regularization. To enhance geometric consistency, we derive a pseudo normal map $\hat{\boldsymbol{n}}$ from the depth map during the optimization process and use it to supervise the rendered normals. Moreover, we incorporate the following total variation loss (TV-loss) from GS-IR to achieve smoother results:

$$\mathcal{L}_n = \mathcal{L}_{n,p} + \lambda_{n-TV} \mathcal{L}_{TV_{\text{normal}}}, \ \mathcal{L}_{n,p} = \|\boldsymbol{n} - \hat{\boldsymbol{n}}\|. \tag{8}$$

## 4.2 DIRECT LIGHTING MODELING

To capture the illumination from various directions and simplify the calculation, we employ image-based lighting (IBL) to model the direct lighting. According to the rendering equation in Eq. 3, we can decompose the outgoing radiation into a diffuse component $L_d$ and a specular component $L_s$. For the diffuse component $L_d$, we store the incident irradiance of direct lighting $I_{dir}$ (see Eq. 7) from different view directions in a prefiltered environment map.

To tackle the intractable integral of the specular component $L_s$, we leverage the widely used split-sum approximation (Karis & Games, 2013) to divide the integral into two parts as follows:

$$\begin{aligned} L_s &= \int_{\Omega} \frac{DFG}{4(\boldsymbol{n} \cdot \boldsymbol{\omega}_i)(\boldsymbol{n} \cdot \boldsymbol{\omega}_o)} L_i(\boldsymbol{\omega}_i, \boldsymbol{x}) (\boldsymbol{\omega}_i \cdot \boldsymbol{n}) d\boldsymbol{\omega}_i \\ &\approx \underbrace{\int_{\Omega} \frac{DFG}{4(\boldsymbol{n} \cdot \boldsymbol{\omega}_o)} d\boldsymbol{\omega}_i}_{\text{BRDF integral } R} \underbrace{\int_{\Omega} L_i(\boldsymbol{\omega}_i) D(\boldsymbol{\omega}_i, \boldsymbol{\omega}_o) (\boldsymbol{\omega}_i \cdot \boldsymbol{n}) d\boldsymbol{\omega}_i}_{\text{Pre-filtered environment map } I_s}. \end{aligned} \tag{9}$$

The first term $R$ represents the integral of BRDF under a constant environment light and is only determined by the angle $\theta$ between the surface normal $\boldsymbol{n}$ and view direction $\boldsymbol{\omega}_o$, as well as the roughness $\rho$. This integral can be precomputed and stored in a 2D look-up table. The second term $I_s$ represents the incident irradiance considering the NDF $D$. Since the NDF $D$ is related to the surface roughness $\rho$ (the rougher the surface, the larger the specular lobe), we can also store the incident irradiance in prefiltered environment maps with different mip-levels for different roughness values. Therefore, the rendering result under direct lighting can be written as:

$$L_{dir} = (1 - m) \frac{\boldsymbol{a}}{\pi} O(\boldsymbol{x}) I_{dir} + R I_s, \tag{10}$$

where $O(\boldsymbol{x})$ is the occlusion at point $\boldsymbol{x}$ obtained through path tracing, as elaborated in Sec. 4.3.

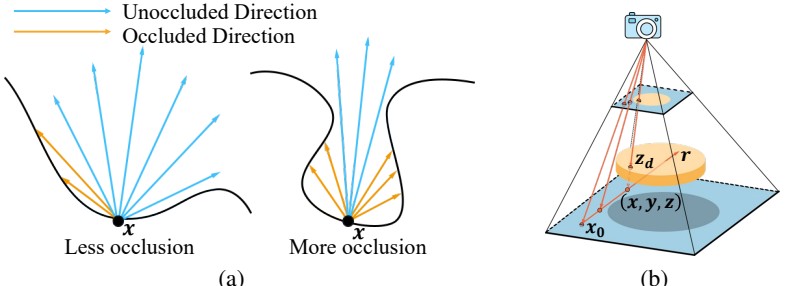

Figure 2: (a) **Occlusion** quantifies the degree to which a surface point can receive ambient light. Points on flat surfaces demonstrate high visibility due to less occlusion from surrounding geometry. In contrast, points located in holes, corners, or adjacent to other surfaces appear darker, since they are more occluded. (b) **Path tracing:** For each surface point corresponding to a pixel on the depth map, we perform ray marching starting from that point to calculate the visibility in different directions.

### 4.3 INDIRECT LIGHTING MODELING

Indirect lighting refers to the illumination that reaches a surface after reflecting off surfaces. It is a crucial component for realistic rendering, particularly in areas occluded by other objects or by the surfaces themselves, as depicted in Fig. 2(a). Typically, to calculate occlusion and indirect lighting, we can sample several rays from the normal-oriented hemisphere $\Omega$ of the surface and determine whether these rays intersect with other surfaces. However, 3DGS explicitly represents scenes as a set of points, which complicates the use of path tracing methods. Inspired by the deferred shading techniques from the gaming industry, we propose a G-buffer based indirect lighting. Our key insight is to recover the geometry from the G-buffer and implement path tracing on the recovered surface. To accelerate the speed of path tracing, we implement a tile-based fast ray marching framework.

**Ambient Occlusion Calculation.** Given the rendered depth map and normal map from the first stage, for a surface point $x = (u, v)$ in the depth map, its position in the view space $(X, Y, Z)$ can be obtained from its depth as $x = (x, y, z)^T = z(u - c_x, v - c_y, 1)^T$ where $z = d(u, v)$ represents its depth and $(c_x, c_y)$ is the coordinate of the optical center. The occlusion at surface point $x$ is expressed as the integral of the visibility function $V$ from various directions $\boldsymbol{\omega} \in \Omega$ as follows:

$$O(\boldsymbol{x}) = 1 - \frac{1}{\pi} \int_{\Omega} V(\boldsymbol{\omega}) \boldsymbol{n} \cdot \boldsymbol{\omega} d\boldsymbol{\omega}, \tag{11}$$

where $V(\boldsymbol{\omega}) = 1$ if a ray starting from point $x$ with direction $\boldsymbol{\omega}$ intersects the surface and $V(\boldsymbol{\omega}) = 0$ otherwise. Since it is difficult to perform uniform sampling directly in solid angle space, we first consider the equivalent form of the original integral in spherical coordinates and uniformly generate sampling vectors to estimate the integral. For more details, please refer to Appendix B.

**Adaptive Path Tracing.** Path tracing is used to determine whether a ray hits the surface. As illustrated in Fig. 2(b), consider a ray starting from a point $x_0 \in \mathbb{R}^{3 \times 1}$ with direction $\boldsymbol{\omega} \in \mathbb{R}^{3 \times 1}$, any point $x = (x, y, z) \in \mathbb{R}^{3 \times 1}$ on this ray can be written as $x = x_0 + \boldsymbol{\omega}t$, where $t$ is the distance from $x_0$. Then, we trace the sample points along the ray. The position of sample point $x$ in the pixel coordinate $(U, V)$ is obtained as $x = (u, v) = \left(\frac{x}{z} - c_x, \frac{y}{z} - c_y\right)$. If $z$ is larger than the depth $z_d = d(u, v)$ at the corresponding coordinate on the depth map and less than $z_d + \delta$, where $\delta$ is the thickness of the surface, the ray intersects a surface and $x$ is occluded in the $\boldsymbol{\omega}$ direction. That is,

$$V(\boldsymbol{\omega}) = \begin{cases} 1 & \text{if } z_d < z < z_d + \delta, \\ 0 & \text{else.} \end{cases} \tag{12}$$

To handle the issue of perspective projection, where the marching of ray in 3D space is not linear to the marching of ray in 2D screen space, we adaptively adjust the step length based on the distance of the current ray-marching starting point. Please refer to Appendix B for more details.

**Indirect Lighting Calculation.** When calculating indirect lighting, other objects occluding the surface point are regarded as indirect lighting sources, as shown in Fig. 3(a). According to rendering equation in Eq. 3, the incoming radiation at point $x$ in direction $\boldsymbol{\omega}$ is the outgoing radiation of the intersection point $\hat{x}$ in direction $-\boldsymbol{\omega}$, i.e., $L_i(\boldsymbol{x}, \boldsymbol{\omega}) = L_o(\hat{\boldsymbol{x}}, -\boldsymbol{\omega})$. To obtain the radiation from intersection points, we first render the RGB image $I_{dir}$ of the scene, considering only direct lighting (see Sec. 4.2). Then, we employ the aforementioned path tracing method to determine if a ray

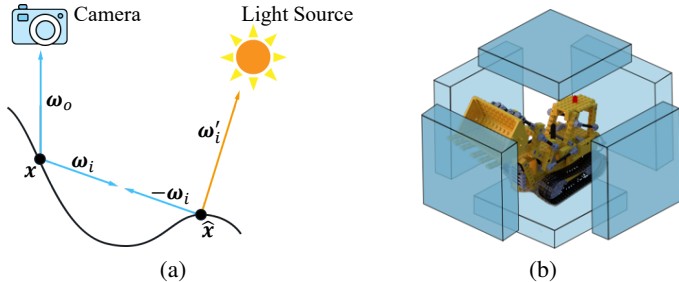

Figure 3: (a) When the surface point $x$ is occluded by $\hat{x}$ in direction $\omega_i$, $\hat{x}$ acts as an indirect light source. In this case, the light received by $x$ in direction $\omega_i$ is equivalent to the light from the actual light source reflected by $\hat{x}$ in the $-\omega_i$ direction. (b) To obtain global geometry and lighting information, we rotate the camera to render the results of the remaining five perspectives and form a cubemap. This cubemap is then utilized to restore the global geometry and facilitate path tracing.

intersects with other surfaces. Once an intersection point $\hat{x}$ is identified, the outgoing radiation is obtained by indexing the corresponding pixel value $I_{dir}(\hat{u}, \hat{v})$ in the RGB image based on the position of the intersection point $(\hat{u}, \hat{v})$. Consequently, the outgoing diffuse radiation under indirect lighting has a similar form to the occlusion integral in Eq. 11 as follows:

$$L_{ind} = (1 - m)\frac{a}{\pi}\int_{\Omega} L_i(\omega_i, x)\, n \cdot \omega_i d\omega_i, \quad L_i(\omega_i, x) = V(\omega_i, x)I_{dir}(\hat{u}, \hat{v}). \qquad (13)$$

We use uniform sampling to calculate indirect lighting and obtain the final rendering result as:

$$L_o(\omega_o, x) = L_{dir} + L_{ind} = \underbrace{(1 - m)\frac{a}{\pi}O(x)I_{dir} + RI_s}_{\text{First-pass}} + \underbrace{L_{ind}}_{\text{Second-pass}}. \qquad (14)$$

To optimize the materials and lighting, we minimize the following decomposition loss:

$$\mathcal{L}_d = \underbrace{\mathcal{L}_1}_{\mathcal{L}_{\text{color}}} + \underbrace{\lambda_M \mathcal{L}_{TV_{\text{mat}}}}_{\mathcal{L}_{\text{material}}} + \underbrace{\lambda_E \mathcal{L}_{TV_{\text{light}}}}_{\mathcal{L}_{\text{light}}}, \qquad (15)$$

where the first term is the color loss, and the remaining two terms are the material TV loss $\mathcal{L}_{\text{material}}$ and lighting TV loss $\mathcal{L}_{\text{light}}$ introduced in GS-IR (Liang et al., 2024) to smooth the materials and lighting. Please refer to Appendix B for more details about $\mathcal{L}_{\text{material}}$ and $\mathcal{L}_{\text{light}}$.

### 4.4 Cubemap based indirect lighting

In Sec. 4.3, the scene geometry is recovered from the depth map. However, this approach only calculates occlusion and indirect lighting in screen space, considering only the local geometry around the surface point. For complex real-world scenes, occlusion and indirect lighting may originate from areas outside the camera's frustum. Thus, it is necessary to model the scene's global geometry. To this end, we extend our path tracing-based indirect lighting from screen space to world space. Specifically, as illustrated in Fig. 3(b), we utilize a cubemap to recover the global geometry and calculate indirect lighting based on it. By taking the current rendering result (depth map, normal map, and RGB image $I_{dir}$) as the front face, we change the camera's pose to render the remaining five faces of the cubemap. Each face of the depth cubemap can be roughly regarded as the geometry of the scene in that direction, and the corresponding indirect illumination is contained in the RGB cubemap. Then, we calculate the indirect lighting by path tracing, as elaborated in Sec. 4.3.

## 5 Experiments

### 5.1 Settings

**Datasets and Metrics.** We perform experiments on the TensoIR synthetic dataset (Jin et al., 2023) and the Mip-NeRF 360 (Mildenhall et al., 2021) dataset. For both datasets, we evaluate the quality of novel view synthesis to compare the overall performance. Notably, the TensoIR dataset provides ground truth results for albedo and relighting under different lighting conditions, which allows us to further evaluate the performance of our estimated albedo and relighting outputs. To quantify the performance, we employ three key metrics: PSNR, SSIM, and LPIPS.

Table 1: **Quantitative results on the TensoIR dataset.** The results show that our method surpasses all baselines in novel view synthesis and albedo reconstruction. It ranks second only to TensoIR in the relighting results. Notably, our method effectively models indirect lighting when relighting.

| | NVS | | | Albedo | | | Relighting | | | Time |
|---|---|---|---|---|---|---|---|---|---|---|
| | PSNR ↑ | SSIM ↑ | LPIPS ↓ | PSNR ↑ | SSIM ↑ | LPIPS ↓ | PSNR ↑ | SSIM ↑ | LPIPS ↓ | |
| NeRFactor | 24.68 | 0.922 | 0.120 | 25.13 | 0.940 | 0.109 | 23.38 | 0.908 | 0.131 | >100 hrs |
| InvRender | 27.37 | 0.934 | 0.089 | 27.34 | 0.933 | 0.100 | 23.97 | 0.901 | 0.101 | 14 hrs |
| NVDiffrec | 30.70 | 0.962 | 0.052 | 29.17 | 0.908 | 0.115 | 19.88 | 0.879 | 0.104 | <1 hr |
| TensoIR | 35.09 | **0.976** | 0.040 | 29.27 | **0.950** | 0.085 | **28.58** | **0.944** | **0.081** | 4.5 hrs |
| GS-IR | 35.33 | 0.974 | 0.039 | 30.29 | 0.941 | **0.084** | 24.37 | 0.885 | 0.096 | 33 min |
| Ours | **36.75** | 0.972 | **0.037** | **31.97** | 0.941 | 0.085 | 24.70 | 0.886 | 0.106 | 28 min |

Figure 4: **Qualitative comparison on the TensoIR dataset.** We visualize the rendering results, reconstructed albedo, and relighting results, respectively.

**Baselines.** We compare our GI-GS method against several baselines that focus on inverse rendering, including NeRFactor (Zhang et al., 2021b), InvRender (Zhang et al., 2022), NVDiffrec (Munkberg et al., 2022), TensoIR (Jin et al., 2023), as well as recent state-of-the-art (SOTA) 3DGS-based methods, GaussianShader (Jiang et al., 2024) and GS-IR (Liang et al., 2024). We also compare our method with some previous methods aimed at novel view synthesis, including NeRF++(Zhang et al., 2020), Instant NGP (Müller et al., 2022), Mip-NeRF360 (Barron et al., 2021), and 3DGS (Kerbl et al., 2023). For quantitative comparisons, we report the results from their original papers, while the results of GaussianShader are from (Du et al., 2024). For qualitative comparisons, we reproduce the results of TensoIR and GS-IR using their released codes.

## 5.2 RESULTS AND COMPARISONS

**TensoIR Dataset.** We first evaluate our approach against previous SOTA methods on the TensoIR dataset. Table 1 presents a quantitative comparison. Our approach outperforms previous methods in both novel view synthesis and albedo estimation in terms of PSNR with the shortest training time, while maintaining comparable SSIM and LPIPS scores. This demonstrates the efficiency and effectiveness of our approach in material estimation and global illumination decomposition. In the relighting results, our method ranks second only to TensoIR and surpasses all other methods. We also visualize the qualitative results in Fig. 4, which demonstrate that our rendering results are closer to the ground truth in overall color accuracy and have a smoother appearance in occluded areas.

**Mip-NeRF 360 Dataset.** Table 2 presents the quantitative results on the Mip-NeRF 360 dataset, and Fig. 5 showcases the rendering outcomes. Our method surpasses previous 3DGS-based inverse rendering approaches across all metrics, achieving superior performance. Notably, our method demonstrates a significant advantage in rendering indoor scenes, where contain more occlusions. This highlights the superiority of our approach in modeling indirect lighting. Moreover, we extend our method from screen space to world space, as elaborated in Sec. 4.4, and refer to the extended version as *Ours-Cubemap*. While this extension incurs a slight decline in quantitative performance, the rendering results reveal more accurate and smoother modeling of occluded areas.

**Indirect Lighting.** To demonstrate the accuracy of our indirect lighting modeling, we separately render the occlusion map and the image under indirect illumination, as shown in Fig. 6. GS-IR uses volumes to store occlusion and indirect lighting. However, during optimization, not all volumes are sampled, particularly those farther from the camera, resulting in missing occlusion and lighting in many areas of the scene. In contrast, our method leverages adaptive path tracing to effectively

Table 2: **Quantitative results on the Mip-NeRF 360 dataset.** Our GI-GS outperforms previous inverse rendering methods and even surpasses some methods designed for novel view synthesis.

| | Outdoor | | | Indoor | | |
| --- | --- | --- | --- | --- | --- | --- |
| | PSNR ↑ | SSIM ↑ | LPIPS ↓ | PSNR ↑ | SSIM ↑ | LPIPS ↓ |
| NeRF | 21.46 | 0.458 | 0.515 | 26.84 | 0.790 | 0.370 |
| Instant NGP | 22.90 | 0.566 | 0.371 | 29.15 | 0.880 | 0.216 |
| Mip-NeRF 360 | 24.47 | 0.691 | 0.283 | 31.72 | 0.917 | 0.180 |
| 3DGS | 24.64 | 0.731 | 0.234 | 30.41 | 0.920 | 0.189 |
| GaussianShader | 22.80 | 0.665 | 0.297 | 26.19 | 0.876 | 0.243 |
| GS-IR | 23.45 | 0.671 | 0.284 | 27.46 | 0.863 | 0.237 |
| Ours | **24.01** | **0.701** | **0.250** | 29.29 | **0.896** | **0.152** |
| Ours-Cubemap | 23.81 | 0.695 | 0.251 | 29.07 | 0.892 | 0.161 |

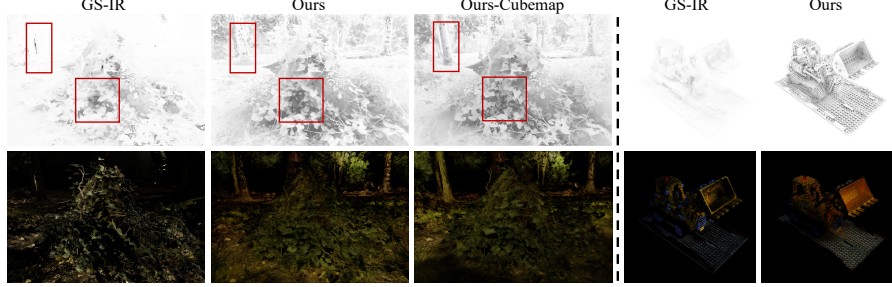

Figure 5: **Qualitative comparison on the Mip-NeRF 360 dataset.** Our GI-GS effectively reconstructs high-frequency details and occluded areas in real-world scenes. Notably, the extended version, Ours-Cubemap, achieves a smoother result in occluded areas.

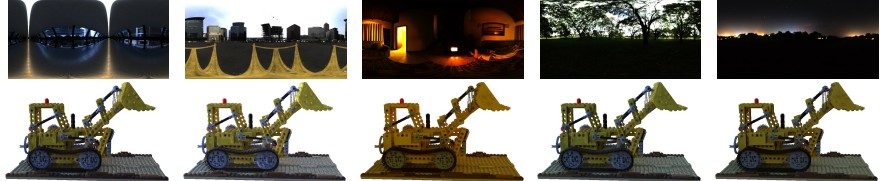

Figure 6: **Qualitative comparison of indirect lighting.** For indirect illumination, we increase the brightness in linear space to make it easier to see the results. Our GI-GS provides more natural and smoother indirect lighting. Moreover, the indirect lighting of GI-GS covers most areas in the scene, especially when extending it to Ours-Cubemap, while the results of GS-IR have large areas missing.

Figure 7: **Relighting visualization** on the TensoIR dataset using different environment maps.

capture occlusion and indirect lighting across most areas. By rendering from multiple perspectives to capture world space information, our extended Ours-Cubemap method achieves smoother and more accurate indirect lighting while maintaining multi-view consistency, especially for distant objects.

**Relighting.** Fig. 7 visualizes the relighting results on the TensoIR dataset under different light conditions. Our method achieves high-quality relighting results, which demonstrates the high accuracy of estimated materials and calculated indirect lighting. More results are provided in the Appendix.

Table 3: **Ablation on occlusion and indirect lighting.** By incorporating both occlusion and indirect lighting, the rendering process more faithfully captures the light and surface interactions.

| | TensoIR | | | Mip-NeRF 360 | | |
|---|---|---|---|---|---|---|
| | PSNR ↑ | SSIM ↑ | LPIPS ↓ | PSNR ↑ | SSIM ↑ | LPIPS ↓ |
| Ours w/o occlusion | 36.37 | 0.970 | 0.041 | 26.24 | 0.788 | 0.210 |
| Ours w/o indirect lighting | 35.60 | 0.965 | 0.047 | 25.89 | 0.778 | 0.212 |
| Ours | **36.75** | **0.972** | **0.037** | **26.35** | **0.788** | **0.206** |

Table 4: **Ablation on the number of sampled rays.** Our training speed remains comparable with GS-IR on the TensoIR dataset and is significantly faster on the Mip-NeRF 360 dataset.

| | TensoIR | | | Time | Mip-NeRF 360 | | | Time |
|---|---|---|---|---|---|---|---|---|
| | PSNR ↑ | SSIM ↑ | LPIPS ↓ | (min) | PSNR ↑ | SSIM ↑ | LPIPS ↓ | (min) |
| GS-IR | 35.33 | 0.974 | 0.039 | 33 | 25.38 | 0.757 | 0.268 | 114 |
| Ours ($N_s = 16$) | 36.25 | 0.966 | 0.047 | **27** | 26.19 | 0.773 | 0.222 | **55** |
| Ours ($N_s = 64$) | 36.75 | 0.972 | 0.037 | 28 | 26.35 | 0.788 | 0.206 | 57 |
| Ours ($N_s = 256$) | **36.82** | **0.976** | **0.034** | 32 | **26.42** | **0.791** | **0.203** | 64 |

Figure 8: **Qualitative comparison of different numbers of ray samples.** The accuracy of indirect lighting increases with the number of ray samples, leading to smoother and less noisy results.

## 5.3 ABLATION STUDY

**Effectiveness of Indirect Lighting.** To evaluate the impact of our path tracing-based occlusion and indirect lighting, we conduct two experiments: one considering only occlusion and the other considering only indirect lighting. The quantitative results presented in Table 3 show that both components significantly affect on the final rendering quality, with indirect lighting being particularly impactful.

**Number of Sampled Rays.** We also conduct experiment to analyze the impact of the number of ray samples $N_s$. The results in Table 4 and Fig. 8 show that increasing the number of sampled rays $N_s$ enhances the rendering quality, especially when $N_s$ is small. Notably, thanks to our efficient path tracing, increasing the number of ray samples only slightly slows down our training speed.

## 6 CONCLUSION

In this paper, we introduce GI-GS, a novel inverse rendering framework based on 3D Gaussian Splatting. To extend 3DGS to the inverse rendering problem, we augment each Gaussian's original attributes by incorporating surface normals and BRDFs, and optimize them through a differentiable PBR pipeline. The key insight of our approach is to combine deferred shading and path tracing to enable accurate global illumination decomposition while ensuring real-time rendering. Our approach successfully achieves high-fidelity geometry and material reconstruction of objects and scenes, as well as effective global illumination decomposition. Extensive experiments demonstrate that GI-GS outperforms previous NeRF-based and 3DGS-based inverse rendering methods in novel view synthesis, achieving comparable relighting with superior computational efficiency.

**Limitations.** Our approach does not consider the specular component of indirect illumination, as it requires complex Monte Carlo sampling and is a long-standing challenge in computer graphics. Moreover, for complex real-world scenes, using an environment map as the direct lighting source may fail to capture the spatially varying lighting in the environment. In addition, our approach relies on high-quality reconstruction of the geometry, which suggests that the accuracy of path tracing can be improved by incorporating additional geometric constraints.

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

## A  IMPLEMENTATION DETAILS

**Training Details.** The first stage of our training process follows the vanilla 3DGS with the addition of the normal loss term. The training iterations are set to 30K, with the same learning rate as the vanilla 3DGS. Besides, we use the improved densification strategy introduced in GOF (Yu et al., 2024) to reduce blurred areas. The optimization of the materials and the lighting takes 10,000 and 5,000 iterations on the Mip-NeRF 360 and TensoIR datasets, respectively. For the BRDF attributes, we adopt the continuous learning rate decay strategy in Plenoxels(Fridovich-Keil et al., 2022), where the initial learning rate is set to 0.05 and decays to a final learning rates of 0.005. All training is conducted on a single NVIDIA A5000 GPU.

**Loss Functions.** For the first stage, we initialize a set of 3D Gaussians using the following loss function:

$$\mathcal{L}_{\text{init}} = \mathcal{L}_1 + \mathcal{L}_{SSIM} + \mathcal{L}_n, \tag{16}$$

where

$$\mathcal{L}_n = \mathcal{L}_{n,p} + \lambda_{n-TV}\mathcal{L}_{TV_{\text{normal}}}, \tag{17}$$

and $\mathcal{L}_{TV_{\text{normal}}}$ is the total variation (TV) loss conditioned by the predicted normal map $\boldsymbol{N}$ and the given reference image $\boldsymbol{I}$ as follows:

$$\mathcal{L}_{TV_{\text{normal}}} = \frac{1}{|\boldsymbol{N}|}\sum_{i,j}\triangle_{ij}^{\boldsymbol{N}}, \tag{18}$$

and

$$\begin{aligned}\triangle_{ij}^{\boldsymbol{N}} = {} & \exp\left(-|\boldsymbol{I}_{i,j} - \boldsymbol{I}_{i-1,j}|\right)\left(\boldsymbol{N}_{i,j} - \boldsymbol{N}_{i-1,j}\right)^2 \\ & + \exp\left(-|\boldsymbol{I}_{i,j} - \boldsymbol{I}_{i,j-1}|\right)\left(\boldsymbol{N}_{i,j} - \boldsymbol{N}_{i,j-1}\right)^2.\end{aligned} \tag{19}$$

For the optimization of the materials and lighting, the decomposition loss $\mathcal{L}_d$ includes $\mathcal{L}_{\text{color}}$, $\mathcal{L}_{TV_{\text{mat}}}$ and $\mathcal{L}_{TV_{\text{light}}}$:

$$\mathcal{L}_d = \underbrace{\mathcal{L}_1}_{\mathcal{L}_{\text{color}}} + \underbrace{\lambda_M\mathcal{L}_{TV_{\text{mat}}}}_{\mathcal{L}_{\text{material}}} + \underbrace{\lambda_E\mathcal{L}_{TV_{\text{light}}}}_{\mathcal{L}_{\text{light}}}, \tag{20}$$

where $\mathcal{L}_{TV_{\text{mat}}}$ is a smooth term similar to $\mathcal{L}_{TV_{\text{normal}}}$. Given the material map $\boldsymbol{M}$, $\mathcal{L}_{TV_{\text{mat}}}$ is defined as:

$$\mathcal{L}_{TV_{\text{mat}}} = \frac{1}{|\boldsymbol{M}|}\sum_{i,j}\triangle_{ij}^{\boldsymbol{M}}, \tag{21}$$

where

$$\begin{aligned}\triangle_{ij}^{\boldsymbol{M}} = {} & \exp\left(-|\boldsymbol{I}_{i,j} - \boldsymbol{I}_{i-1,j}|\right)\left(\boldsymbol{M}_{i,j} - \boldsymbol{M}_{i-1,j}\right)^2 \\ & + \exp\left(-|\boldsymbol{I}_{i,j} - \boldsymbol{I}_{i,j-1}|\right)\left(\boldsymbol{M}_{i,j} - \boldsymbol{M}_{i,j-1}\right)^2.\end{aligned} \tag{22}$$

Besides, $\mathcal{L}_{TV_{\text{light}}}$ is used in Eq. 20 to obtain a smoother environment light, which is defined as:

$$TV_{\text{light}} = \frac{1}{|\boldsymbol{E}|}\sum_{i,j}\triangle_{ij}^{\boldsymbol{E}}, \tag{23}$$

where

$$\triangle_{ij}^{\boldsymbol{E}} = \left(\boldsymbol{E}_{i,j} - \boldsymbol{E}_{i-1,j}\right)^2 + \left(\boldsymbol{E}_{i,j} - \boldsymbol{E}_{i,j-1}\right)^2, \tag{24}$$

and $\boldsymbol{E}$ is a learnable environment light map. Following GS-IR, we set $\lambda_{n-TV}$, $\lambda_M$, and $\lambda_E$ to 5.0, 1.0, and 0.01, respectively.

## B  TECHNICAL DETAILS

**Pseudo Normal.** We derive the pseudo normals from the depth map $d$. Given a point $d(u, v)$ in the depth map, we consider the local $3 \times 3$ window of depth values centered on this point and project these 9 depth points into the correpsonding 3D points in the scene. By calculating the tangent vectors between these 3D points, we can derive the normals through the cross-product of these tangent vectors. The final pseudo normal for the central pixel corresponding to $(u, v)$ is computed as the average of all these derived normals.

**Uniform Sampling.** Since it is difficult to perform uniform sampling directly in solid angle space, we first consider the equivalent form of the original integral (see Eq. 11) in spherical coordinates as follows:

$$O(\boldsymbol{x}) = 1 - \frac{1}{\pi} \int_{\phi=0}^{2\pi} \int_{\theta=0}^{\frac{\pi}{2}} V(\phi, \theta) \cos(\theta) \sin(\theta) d\phi d\theta. \tag{25}$$

To estimate the integral, we uniformly generate sampling vectors spread inside the normal-oriented hemisphere by dividing each spherical coordinate into $n_1$ and $n_2$ samples to calculate

$$O(\boldsymbol{x}) = 1 - \frac{\sum_{i=0}^{n_1} \sum_{j=0}^{n_2} V(\phi_i, \theta_j) \cos(\theta_j) \sin(\theta_j)}{\sum_{i=0}^{n_1} \sum_{j=0}^{n_2} \cos(\theta_j) \sin(\theta_j)}. \tag{26}$$

Moreover, some previous inverse rendering works, such as NeILF (Yao et al., 2022), use Fibonacci sampling over the hemisphere to generate uniform sampling points on the sphere, which is effective and easy to implement.

**Adaptive Ray Tracing.** Due to the perspective projection, the marching of a ray in 3D space is not linear to the marching of the ray in 2D screen space. For areas of scene that are far from the image plane, traversing a significant distance in 3D may only cover a few pixels on the screen. This can lead to poor sampling accuracy, as we are oversampling several pixels. To address this issue, we adaptively adjust the step length based on the distance of the current ray-marching starting point. Considering a ray that originates from a surface point $\boldsymbol{x_0} = (x_0, y_0, z_0) \in \mathbb{R}^{3\times 1}$ with direction $\boldsymbol{\omega} \in \mathbb{R}^{3\times 1}$, the proposed adaptive ray tracing is expressed as follows:

$$\boldsymbol{x} = \boldsymbol{x_0} + \boldsymbol{\omega} k t, \; k \in \mathbb{Z}, \tag{27}$$

where

$$t = t_0 \left(1 + \frac{z_0}{z_{\text{far}} - z_{\text{near}}}\right)^2. \tag{28}$$

Here, $t_0$ is the pre-defined step length, $z_{\text{far}}$ is the distance from the camera to the far clipping plane, and $z_{\text{near}}$ is the distance to the near clipping plane. This adaptive approach enhances the accuracy of ray tracing by ensuring that step sizes are more appropriate for varying distances in the scene.

## C    ADDITIONAL RESULTS ON THE MIP-NERF 360 DATASET

We provide the quantitative results of novel view synthesis on the Mip-NeRF 360 dataset for each scene, as detailed in Tables 5 – 7. Our method outperforms all the previous 3DGS-based inverse rendering approaches and even some previous methods that focus on novel view synthesis. Importantly, our method achieves SOTA performance in terms of PSNR among all 3DGS-based inverse rendering approaches across all scenes, except for a negligible decline in the garden scene. In addition, we visualize the rendering results, decomposition results (normal, occlusion, indirect illumination), and relighting results in Fig. 9. Notably, our calculated occlusion and indirect illumination model the shadow effect and the bounces of lights across the scene with high fidelity. We also provide more qualitative comparison results with the previous SOTA 3DGS-based baseline GS-IR in Fig. 10 and Fig. 11.

Table 5: PSNR scores for Mip-NeRF360 scenes.

| method | bicycle | flowers | garden | stump | treehill | room | counter | kitchen | bonsai |
|---|---|---|---|---|---|---|---|---|---|
| NeRF++ | 22.64 | 20.31 | 24.32 | 24.34 | 22.20 | 28.87 | 26.38 | 27.80 | 29.15 |
| Plenoxels | 21.91 | 20.10 | 23.49 | 20.66 | 22.25 | 27.59 | 23.62 | 23.42 | 24.67 |
| INGP-Base | 22.19 | 20.35 | 24.60 | 23.63 | 22.36 | 29.27 | 26.44 | 28.55 | 30.34 |
| INGP-Big | 22.17 | 20.65 | 25.07 | 23.47 | 22.37 | 29.69 | 26.69 | 29.48 | 30.69 |
| Mip-NeRF 360 | 24.37 | **21.73** | 26.98 | 26.40 | **22.87** | **31.63** | **29.55** | **32.23** | **33.46** |
| 3DGS | **25.25** | 21.52 | **27.41** | **26.55** | 22.49 | 30.63 | 28.70 | 30.32 | 31.98 |
| GaussianShader | 23.12 | 20.34 | 26.44 | 23.92 | 20.17 | 24.27 | 26.35 | 27.72 | 28.09 |
| GS-IR | 23.80 | 20.57 | 25.72 | 25.37 | 21.79 | 28.79 | 26.22 | 27.99 | 28.18 |
| Ours | 24.32 | 20.90 | 26.42 | 26.08 | 22.31 | 29.89 | 27.18 | 29.94 | 30.15 |
| Ours-Cubemap | 24.21 | 20.85 | 26.05 | 25.89 | 22.05 | 29.60 | 27.03 | 29.60 | 30.05 |

Table 6: SSIM scores for Mip-NeRF360 scenes.

| method | bicycle | flowers | garden | stump | treehill | room | counter | kitchen | bonsai |
|---|---|---|---|---|---|---|---|---|---|
| NeRF++ | 0.526 | 0.453 | 0.635 | 0.594 | 0.530 | 0.530 | 0.802 | 0.816 | 0.876 |
| Plenoxels | 0.496 | 0.431 | 0.606 | 0.523 | 0.509 | 0.842 | 0.759 | 0.648 | 0.814 |
| INGP-Base | 0.491 | 0.450 | 0.649 | 0.574 | 0.518 | 0.855 | 0.798 | 0.818 | 0.890 |
| INGP-Big | 0.512 | 0.486 | 0.701 | 0.594 | 0.542 | 0.871 | 0.817 | 0.858 | 0.906 |
| Mip-NeRF 360 | 0.685 | 0.583 | 0.813 | 0.744 | 0.632 | 0.913 | 0.894 | 0.920 | **0.941** |
| 3DGS | **0.771** | **0.605** | **0.868** | **0.775** | **0.638** | **0.914** | **0.905** | **0.922** | 0.938 |
| GaussianShader | 0.700 | 0.542 | 0.842 | 0.667 | 0.572 | 0.847 | 0.874 | 0.887 | 0.893 |
| GS-IR | 0.706 | 0.543 | 0.804 | 0.716 | 0.586 | 0.867 | 0.839 | 0.867 | 0.883 |
| Ours | 0.740 | 0.574 | 0.830 | 0.751 | 0.608 | 0.901 | 0.860 | 0.908 | 0.917 |
| Ours-Cubemap | 0.737 | 0.574 | 0.817 | 0.749 | 0.598 | 0.892 | 0.857 | 0.901 | 0.917 |

Table 7: LPIPS scores for Mip-NeRF360 scenes.

| method | bicycle | flowers | garden | stump | treehill | room | counter | kitchen | bonsai |
|---|---|---|---|---|---|---|---|---|---|
| NeRF++ | 0.455 | 0.466 | 0.331 | 0.416 | 0.466 | 0.335 | 0.351 | 0.260 | 0.291 |
| Plenoxels | 0.506 | 0.521 | 0.386 | 0.503 | 0.540 | 0.419 | 0.441 | 0.447 | 0.398 |
| INGP-Base | 0.487 | 0.481 | 0.312 | 0.450 | 0.489 | 0.301 | 0.342 | 0.254 | 0.227 |
| INGP-Big | 0.446 | 0.441 | 0.257 | 0.421 | 0.450 | 0.261 | 0.306 | 0.195 | 0.205 |
| Mip-NeRF 360 | 0.301 | 0.344 | 0.170 | 0.261 | 0.339 | 0.211 | 0.204 | 0.127 | 0.176 |
| 3DGS | **0.205** | 0.336 | **0.103** | **0.210** | **0.317** | 0.220 | 0.204 | 0.129 | 0.205 |
| GaussianShader | 0.274 | 0.377 | 0.130 | 0.297 | 0.406 | 0.304 | 0.242 | 0.167 | 0.257 |
| GS-IR | 0.259 | 0.371 | 0.158 | 0.258 | 0.372 | 0.279 | 0.260 | 0.188 | 0.264 |
| Ours | 0.213 | 0.313 | 0.140 | 0.222 | 0.363 | **0.175** | **0.172** | **0.111** | **0.150** |
| Ours-Cubemap | 0.213 | **0.310** | 0.149 | 0.224 | 0.360 | 0.191 | 0.179 | 0.120 | 0.152 |

Table 8: Per-scene results on the TensoIR synthetic dataset. For albedo reconstruction results, we follow NeRFactor (Zhang et al., 2021b) and scale each RGB channel by a global scalar.

| Scene | Method | NVS | | | Albedo | | | Relighting | | |
|---|---|---|---|---|---|---|---|---|---|---|
| | | PSNR ↑ | SSIM ↑ | LPIPS ↓ | PSNR ↑ | SSIM ↑ | LPIPS ↓ | PSNR ↑ | SSIM ↑ | LPIPS ↓ |
| Lego | 3DGS | 39.68 | 0.985 | 0.015 | N/A | N/A | N/A | N/A | N/A | N/A |
| | NeRFactor | 26.08 | 0.881 | 0.151 | **25.44** | **0.937** | **0.112** | 23.25 | 0.865 | 0.156 |
| | InvRender | 24.39 | 0.883 | 0.151 | 21.43 | 0.882 | 0.160 | 20.12 | 0.832 | 0.171 |
| | NVDiffrec | 30.06 | 0.945 | 0.059 | 21.35 | 0.849 | 0.166 | 20.09 | 0.844 | 0.114 |
| | TensoIR | 34.70 | **0.968** | 0.037 | 25.24 | 0.900 | 0.145 | **27.60** | **0.922** | **0.095** |
| | GS-IR | 34.38 | **0.968** | 0.036 | 24.96 | 0.889 | **0.143** | 23.26 | 0.842 | 0.117 |
| | Ours | **35.74** | 0.966 | **0.031** | 24.68 | 0.883 | 0.148 | 23.42 | 0.843 | 0.119 |
| Hotdog | 3DGS | 39.99 | 0.986 | 0.017 | N/A | N/A | N/A | N/A | N/A | N/A |
| | NeRFactor | 24.50 | 0.940 | 0.141 | 24.65 | 0.950 | 0.142 | 22.71 | 0.914 | 0.159 |
| | InvRender | 31.83 | 0.952 | 0.089 | 27.03 | 0.950 | 0.094 | 27.63 | 0.928 | **0.089** |
| | NVDiffrec | 34.90 | 0.972 | 0.054 | 26.06 | 0.920 | 0.116 | 19.07 | 0.885 | 0.118 |
| | TensoIR | **36.82** | **0.976** | 0.045 | **30.37** | **0.947** | 0.093 | **27.93** | **0.933** | 0.115 |
| | GS-IR | 34.12 | 0.972 | 0.049 | 26.75 | 0.941 | **0.088** | 21.57 | 0.888 | 0.140 |
| | Ours | 35.77 | 0.968 | 0.056 | 26.59 | 0.937 | 0.094 | 21.62 | 0.889 | 0.144 |
| Armadillo | 3DGS | 46.50 | 0.991 | 0.023 | N/A | N/A | N/A | N/A | N/A | N/A |
| | NeRFactor | 26.48 | 0.947 | 0.095 | 28.01 | 0.946 | 0.096 | 26.89 | 0.944 | 0.102 |
| | InvRender | 31.12 | 0.968 | 0.057 | 35.57 | 0.959 | 0.076 | 27.81 | 0.949 | 0.069 |
| | NVDiffrec | 33.66 | 0.983 | **0.031** | 38.84 | 0.969 | 0.076 | 23.10 | 0.921 | 0.063 |
| | TensoIR | 39.05 | **0.986** | 0.039 | 34.36 | **0.989** | 0.059 | **34.50** | **0.975** | **0.045** |
| | GS-IR | 39.29 | 0.980 | 0.039 | 38.57 | 0.986 | **0.051** | 27.74 | 0.918 | 0.091 |
| | Ours | **40.37** | 0.980 | 0.036 | **43.71** | 0.983 | 0.053 | 28.10 | 0.917 | 0.080 |
| Ficus | 3DGS | 41.11 | 0.994 | 0.007 | N/A | N/A | N/A | N/A | N/A | N/A |
| | NeRFactor | 21.66 | 0.919 | 0.095 | 22.40 | 0.928 | 0.085 | 20.68 | 0.907 | 0.107 |
| | InvRender | 22.13 | 0.934 | 0.057 | 25.33 | 0.942 | 0.072 | 20.33 | 0.895 | 0.073 |
| | NVDiffrec | 22.13 | 0.946 | 0.064 | 30.44 | 0.894 | 0.101 | 17.26 | 0.865 | 0.073 |
| | TensoIR | 29.78 | 0.973 | 0.041 | 27.13 | **0.964** | **0.044** | 24.30 | **0.947** | **0.068** |
| | GS-IR | 33.55 | **0.976** | 0.031 | 30.87 | 0.948 | 0.053 | 24.93 | 0.893 | 0.081 |
| | Ours | **35.13** | **0.976** | **0.023** | **32.90** | 0.961 | 0.047 | **25.65** | 0.895 | 0.081 |

# D  ADDITIONAL RESULTS ON THE TENSOIR SYNTHETIC DATASET

We provide the quantitative results of novel view synthesis, albedo estimation, and relighting on the TensoIR synthetic dataset for each object, as reported in Table 8. Our method outperforms all previ-

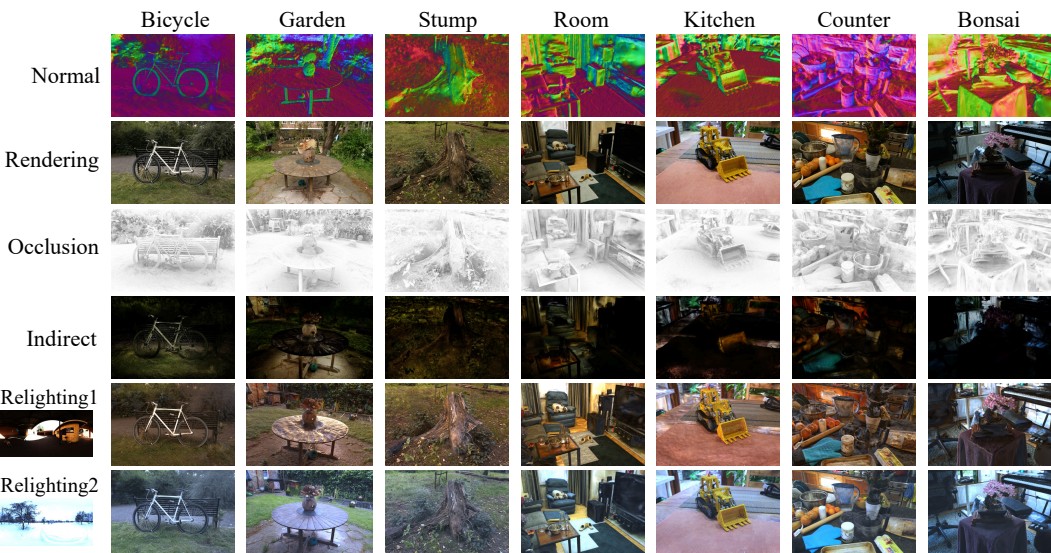

Figure 9: Visualization of our inverse rendering, indirect lighting and relighting results on the Mip-NeRF 360 dataset.

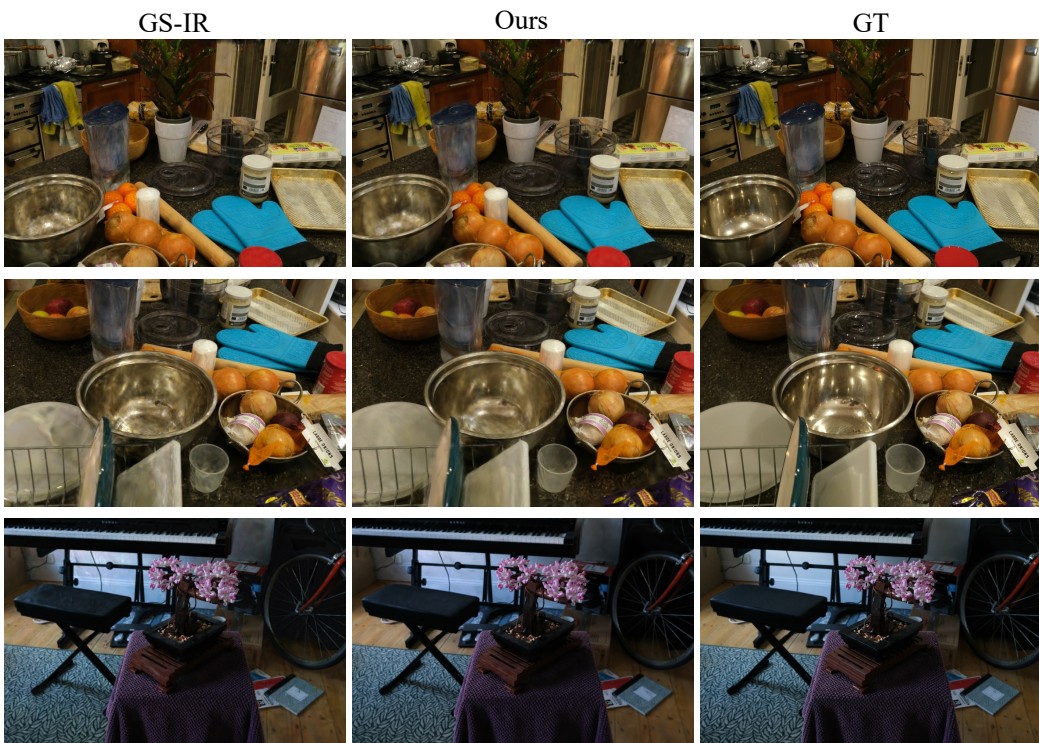

Figure 10: More qualitative comparison results on the Mip-NeRF 360 dataset.

ous inverse rendering approaches in novel view synthesis. For albedo estimation and relighting, our method surpasses the previous SOTA 3DGS-based baseline GS-IR. Besides, we visualize the rendering results, decomposition results (normal, albedo, roughness, occlusion, indirect illumination), and relighting results in Fig. 12. Moreover, Fig. 13 provides the material editing results on the TensoIR dataset.

GS-IR                    Ours                    GT

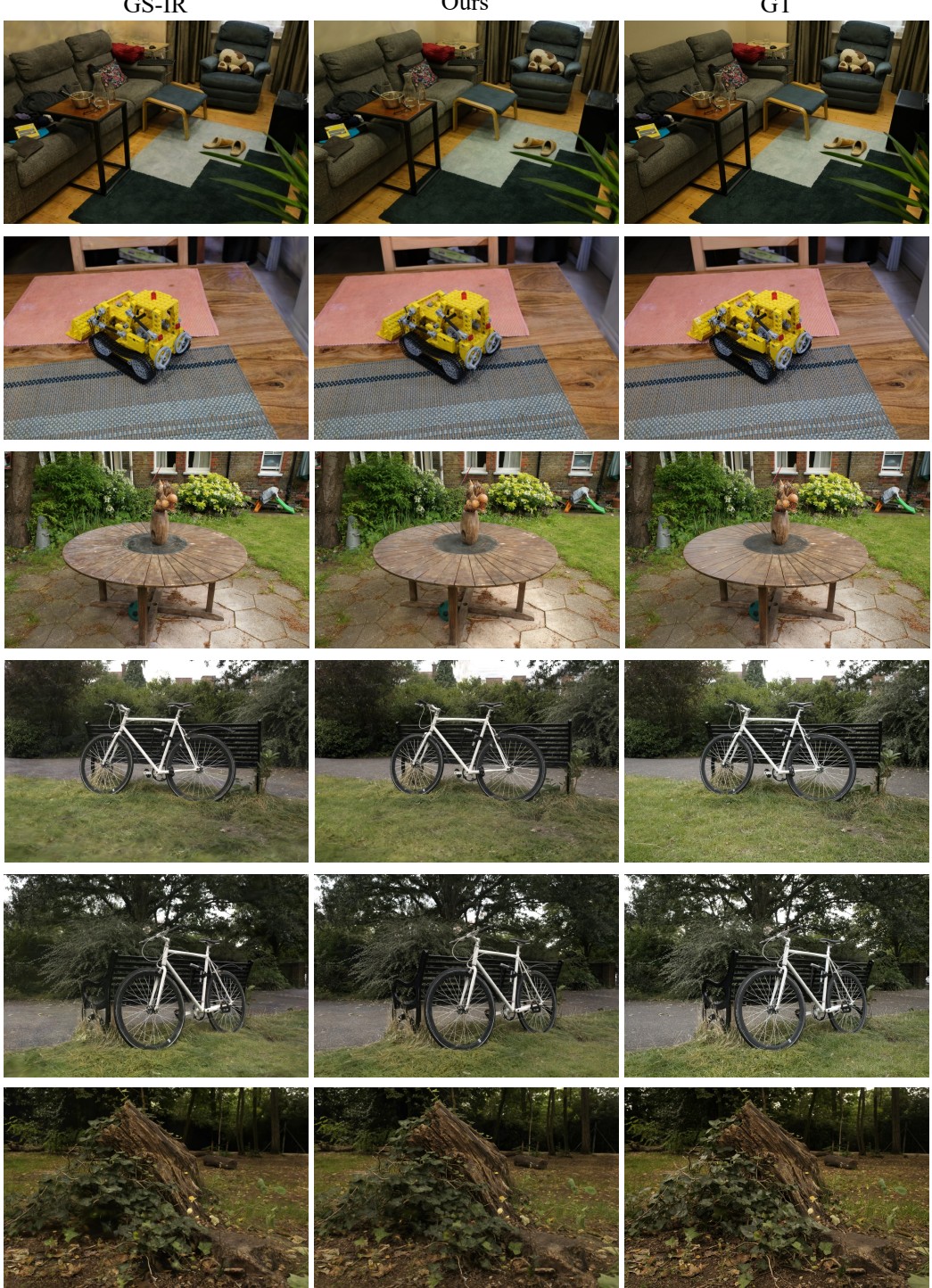

Figure 11: More qualitative comparison results on the Mip-NeRF 360 dataset.

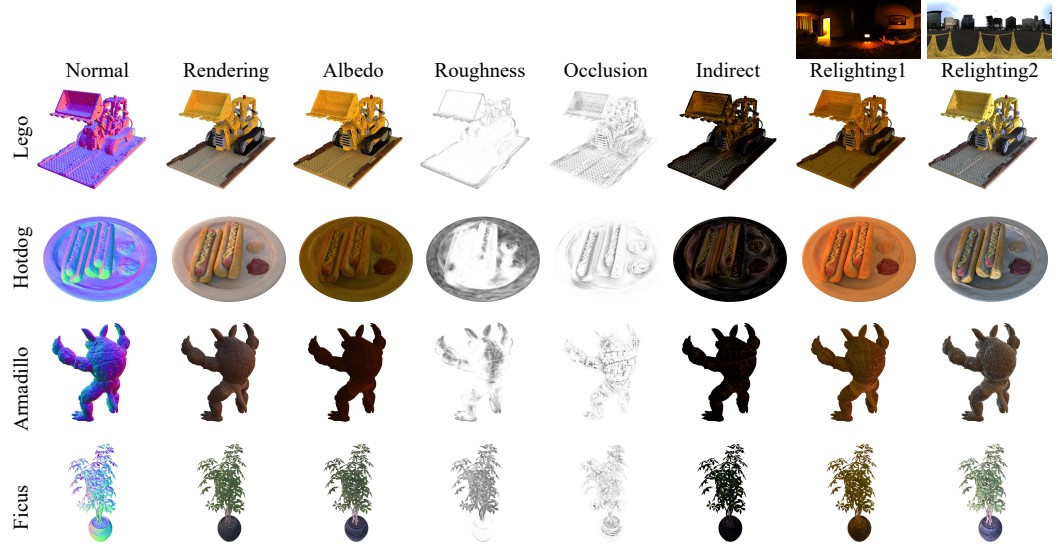

Figure 12: Visualization of our inverse rendering, indirect lighting and relighting results on the TensoIR synthetic dataset.

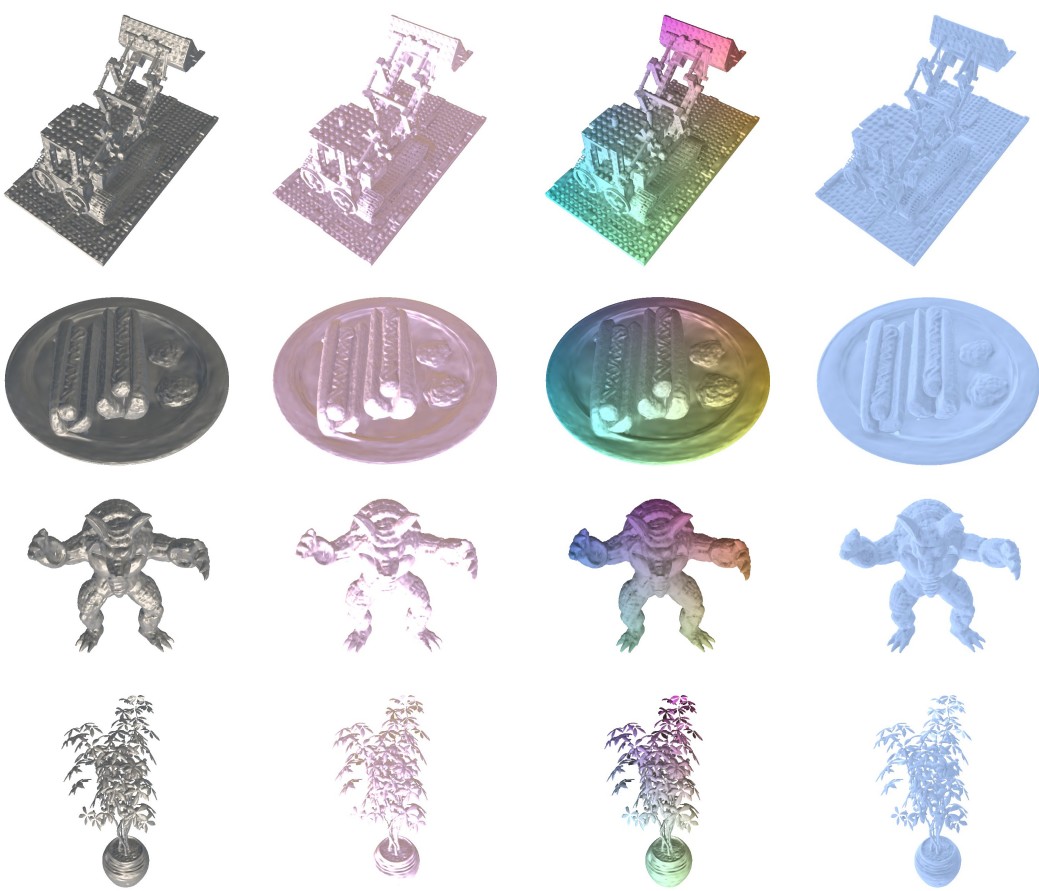

Figure 13: Visualization of material editing on the TensoIR synthetic dataset.

# E    ANALYSIS OF NORMAL ESTIMATION

In this section, we explore the impact of normal estimation quality on relighting. The accuracy of normals is crucial to the quality of PBR results. However, our method does not introduce additional supervision on normals and relies only on the geometry from the vanilla 3DGS. Previous experimental results show that inadequate normal estimation will greatly degrade the quality of relighting, because the split-sum approximation relies on the normals to determine lighting at shading points. To verify this viewpoint, we use the relatively accurate normals predicted by the Omnidata model (Eftekhar et al., 2021) to supervise the rendered normals, which provides additional geometric priors. As illustrated in Fig. 14, with the guidance of Omnidata model, the quality of normal estimation is greatly improved. Consequently, the relighting results are also significantly enhanced, thanks to more accurate normal estimation, as shown in Fig. 15.

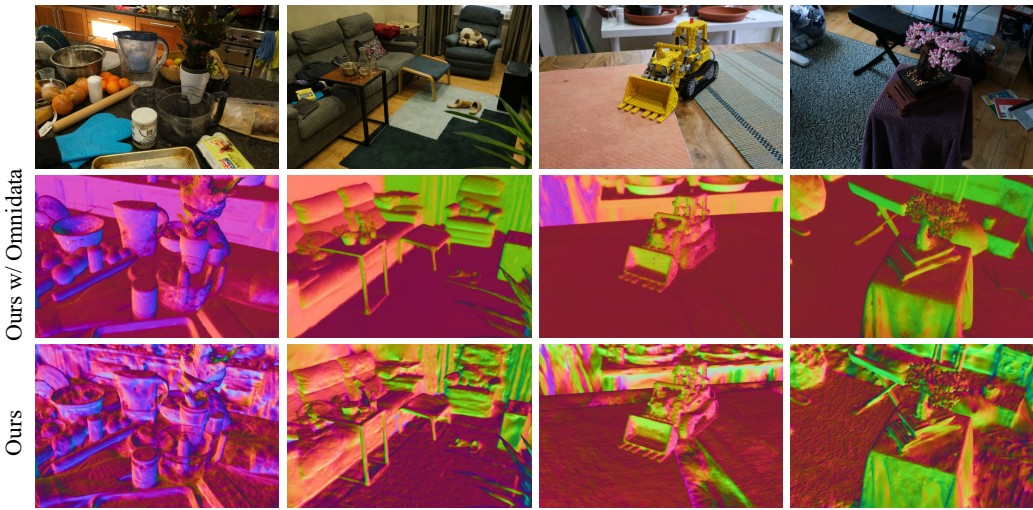

Figure 14: Qualitative comparison of normal estimation before and after the supervision by the Omnidata model.

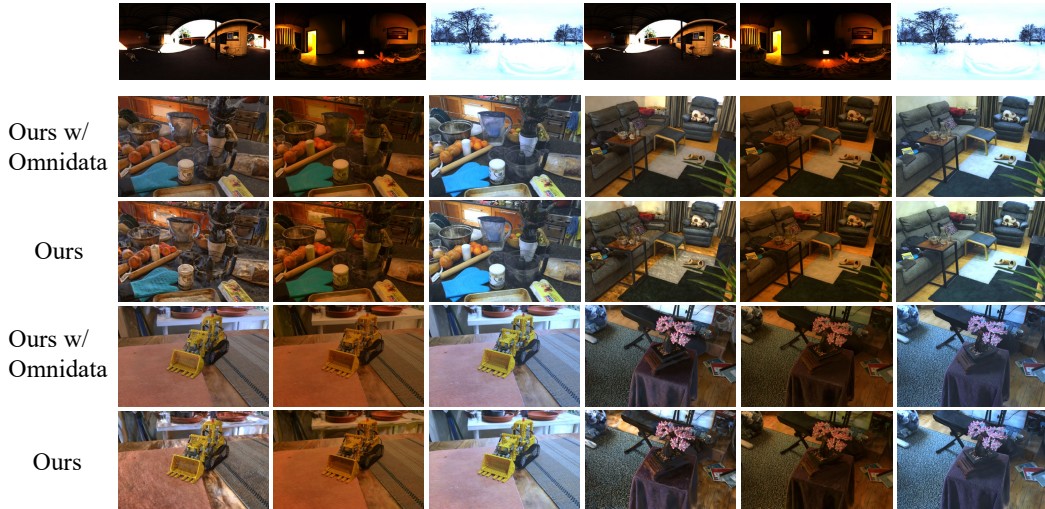

Figure 15: Qualitative comparison of relighting before and after the supervision by the Omnidata model.

## F ANALYSIS OF GLOBAL ILLUMINATION

In this section, we evaluate the quality, controllability, and impact of global illumination on the final rendering result. We also add some implementation details of global illumination.

### F.1 COMPARISON ON OCCLUSION

We compare the quality of estimated occlusion between our method and two baselines. As illustrated in Fig. 16, our method accurately estimates occlusion at both object and scene levels. In contrast, GS-IR faces challenges in geometrically complex regions and struggles to accurately reconstruct the occlusion of distant objects at the scene level, often leading to incorrect estimations of occlusion in certain areas. Although TensoIR estimates high-quality occlusion, it is limited to the object level and cannot be applied to scene-level situations. Besides, for unoccluded areas, TensoIR may produce some degree of occlusion. This is likely because the visibility of TensoIR is estimated by an MLP instead of querying whether the ray intersects the surface.

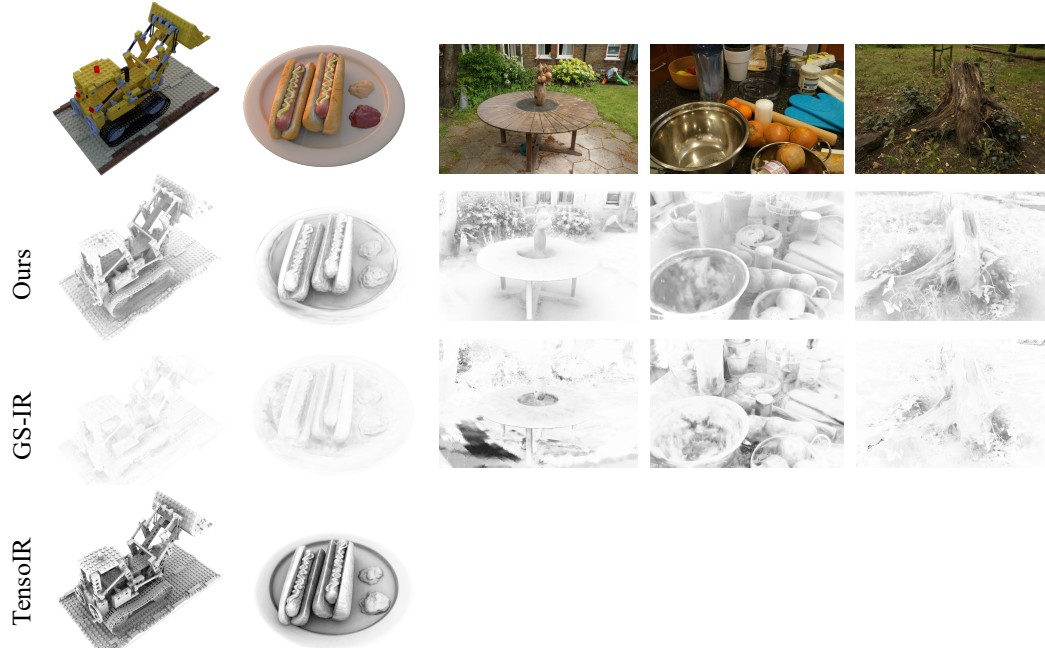

Figure 16: Qualitative comparison of occlusion.

### F.2 INDIRECT ILLUMINATION CALCULATION

In Sec. 4.3, we introduce how to calculate indirect illumination from the first-pass rendering result. In this section, We will explain that this method of calculating indirect illumination is a feasible approximation with high efficiency. As illustrated in Fig. 17, for a surface point $x$, the indirect lighting $L_i(x, \omega)$ in direction $\omega$ is equal to the outgoing radiance $L_o(\hat{x}, -\omega)$ of surface point $\hat{x}$ in direction $-\omega$. Besides, the RGB value $I(u, v)$ corresponding to $\hat{x}$ in the first-pass rendering result is equal to the outgoing radiance $L_o(\hat{x}, \omega_1)$ in direction $\omega_1$. Since the diffuse component has no directionality, the difference between $L_o(\hat{x}, -\omega)$ and $L_o(\hat{x}, \omega_1)$ is the specular component:

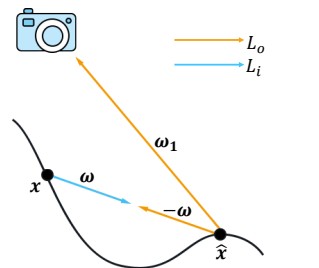

Figure 17: Indirect illumination calculation.

$$
\begin{aligned}
\Delta L_o &= L_o(\hat{x}, -\omega) - L_o(\hat{x}, \omega_1) \\
&= (L_d(\hat{x}, -\omega) + L_s(\hat{x}, -\omega)) - (L_d(\hat{x}, \omega_1) + L_s(\hat{x}, \omega_1)) \\
&= L_s(\hat{x}, -\omega) - L_s(\hat{x}, \omega_1).
\end{aligned}
\tag{29}
$$

In most practical scenarios, the specular component $\Delta L_s$ is relatively negligible compared to the diffuse component. Therefore, we believe that substituting $I(u, v)$ to replace $L_o(\hat{x}, -\omega)$ serves as a practical approximation that avoids the need for additional calculations. The visualization results in Fig. 19 also support this claim, where the tone and details of indirect illumination are satisfactory.

## F.3 THICKNESS IN PATH TRACING

In this section, we provide a detailed explanation of the thickness parameter $\delta$ introduced in Eq. 12. Consider a scenario where a ray does not intersect with a surface. If we compare the depth $z$ of the sampling point to the corresponding value $z_d$ in the depth map, a situation arises where $z_d < z$. In this case, the ray would incorrectly be treated as intersecting with the surface, as illustrated on the left side of Fig. 18.

With the introduction of the thickness parameter $\delta$, we refine this condition. As shown in the middle of Fig. 18, if $z$ falls outside the range defined by $z_d$ and $z_d + \delta$, the ray does not satisfy the criteria for intersection with the surface. Consequently, we conclude that the ray does not intersect with the surface. Only when the depth of the sampling point $z$ lies within the interval $[z_d, z_d + \delta]$, as depicted on the right side of Fig. 18, we consider the ray to be intersecting with the surface.

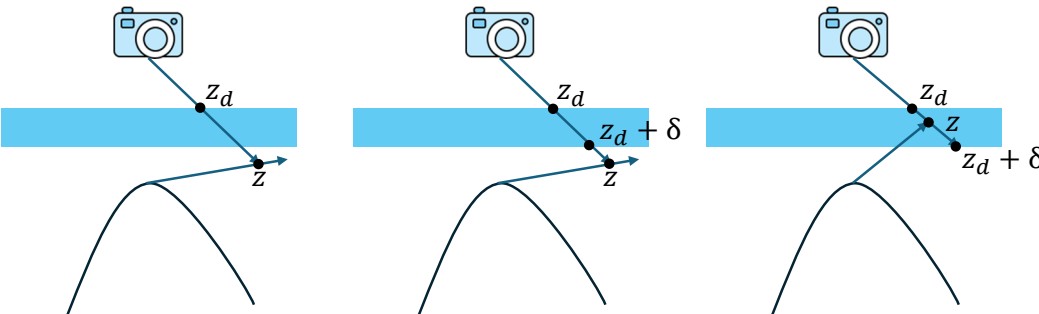

Figure 18: Illustration of the determination of intersection.

## F.4 CONTROLLABILITY OF GLOBAL ILLUMINATION

In this section, we demonstrate the impact of different path tracing settings on global illumination. As illustrated in Fig. 19, changing the path tracing parameters, such as the step size and the number of steps produces different global illumination results can be obtained. This controllability allows our method to accommodate a wide range of objects and scenes.

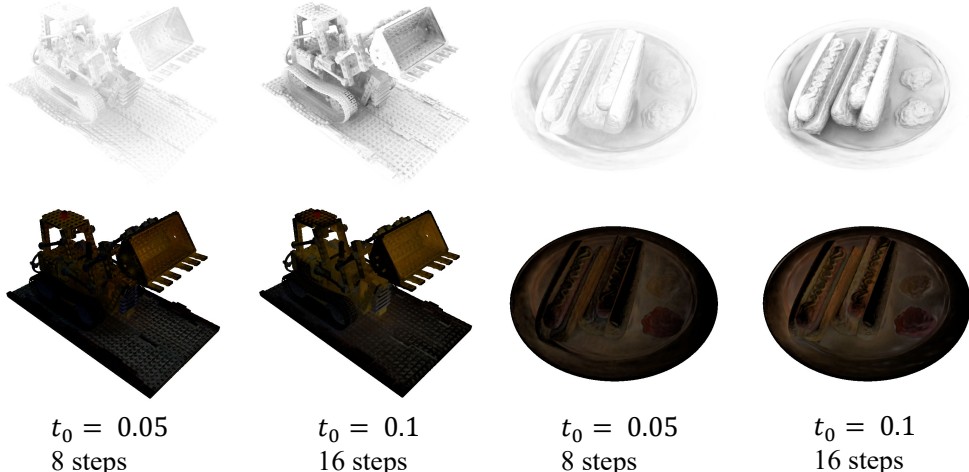

| $t_0 = 0.05$ | $t_0 = 0.1$ | $t_0 = 0.05$ | $t_0 = 0.1$ |
| 8 steps | 16 steps | 8 steps | 16 steps |

Figure 19: Visualization of occlusion and indirect illumination under different settings.

### F.5 SHARP SHADOW UNDER NATURAL LIGHT SOURCE

The occlusion in our method is obtained by integrating the visibility function over the hemisphere. Therefore, for natural light sources with directionality, this method struggles to accurately approximate sharp shadows. To address this issue, we can employ shadow map, which is widely used in computer graphics, to obtain accurate shadows. Specifically, by rendering a depth map at the location of the light source and comparing the depth of the shaded point relative to the light source and the value in the depth map, it is possible to determine whether the shaded point is in shadow. In Fig. 20, we present the result of relighting under a point light using the shadow map, from which sharp shadows can be observed.

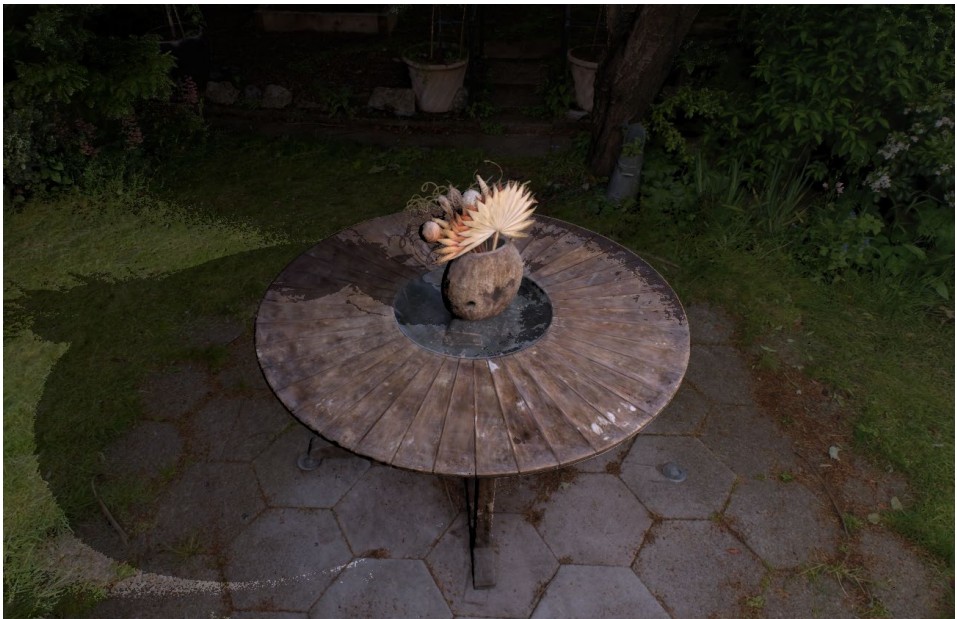

Figure 20: Relighting under point light source with shadow map.

## G  LIGHTING DECOMPOSITION

In this section, we visualize the different components in the rendering result. The decomposed diffuse and specular components are illustrated in Fig. 21. It can be seen that our method successfully decouples the diffuse and specular components. Besides, in the top of Fig. 22, we visualize the direct lighting during relighting, from which we can see the importance of accurate normal estimation for the quality of direct lighting. Moreover, at the bottom of Fig. 22, we show the contribution of indirect lighting to the rendering results, which is especially important for the correct rendering of occluded areas.

## H  ABLATION STUDY ON INVERSE RENDERING PERFORMANCE

To validate the effectiveness of the proposed path tracing-based occlusion and indirect illumination, we compare the performance of material estimation and relighting in three settings: with both occlusion and indirect illumination, with occlusion only, and with indirect illumination only. The quantitative results in Table 9 show that both occlusion and indirect illumination have a significant impact on the performance of inverse rendering. In addition, we provide a qualitative comparison in Fig. 23. The results indicate that the absence of either occlusion or indirect illumination leads to degradation in reconstruction quality of occluded areas. Specifically, without the help of occlusion, the rendering of occluded areas tends to be brighter than the actual scene, leading to more shadows in the estimated albedo. Moreover, without the compensation from indirect illumination, occluded areas appear darker and bright spots emerge in the albedo.

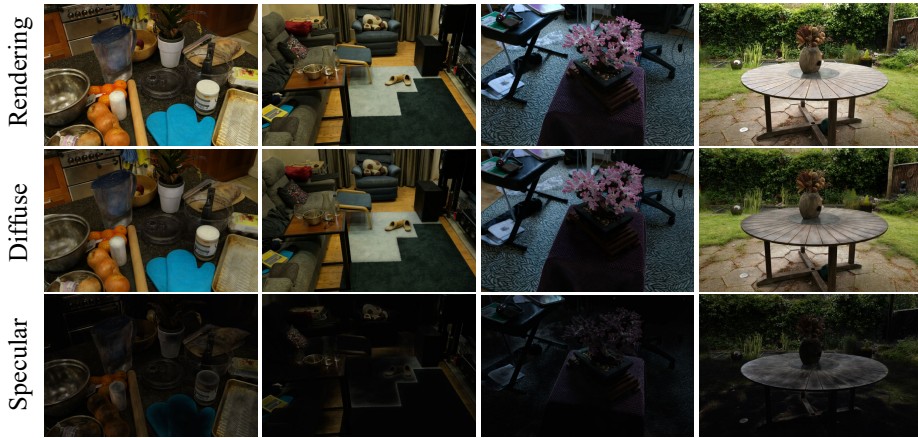

Figure 21: Visualization of decomposed diffuse and specular components.

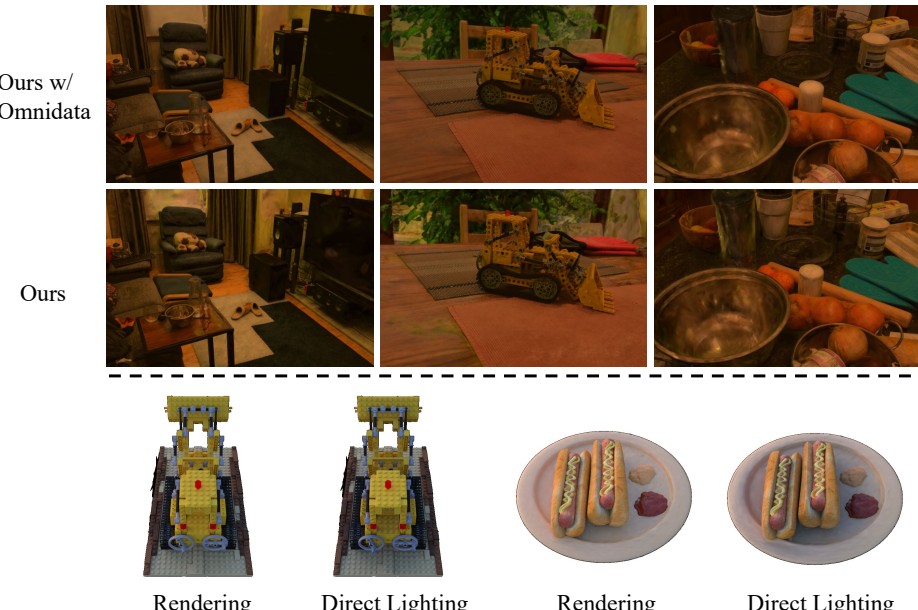

Figure 22: **Top**: Direct lighting during relighting. **Bottom**: Comparison between the rendering result and only direct lighting.

# I  MORE RESULTS

In this section, we provide additional experiment results on the Stanford-ORB (Kuang et al., 2024) and Synthetic4Relight (Zhang et al., 2022) datasets.

## I.1  RESULTS ON THE SYNTHETIC4RELIGHT DATASET

To further compare the novel-view synthesis, material estimation, and relighting performance, we conduct experiments on the Synthetic4Relight dataset (Zhang et al., 2022). Table 10 presents a quantitative comparison. Our approach outperforms previous methods in both novel view synthesis and relighting, achieving higher PSNR and lower LPIPS scores while maintaining comparable SSIM. In the material estimation results, our method is only slightly inferior to InvRender (Zhang et al., 2022) and surpasses all other baselines. We also provide qualitative results in Fig. 24, Fig. 25, and Fig. 26, which demonstrate that our method achieves more accurate material estimation and relighting results than the previous GS-based inverse rendering method.

Table 9: Ablation on material estimation and relighting.

|  | Albedo | | | Relighting | | |
|---|---|---|---|---|---|---|
|  | PSNR ↑ | SSIM ↑ | LPIPS ↓ | PSNR ↑ | SSIM ↑ | LPIPS ↓ |
| Ours w/o occlusion | 31.39 | 0.928 | 0.093 | 24.38 | 0.879 | 0.111 |
| Ours w/o indirct lighting | 31.46 | 0.929 | 0.089 | 24.30 | 0.877 | 0.113 |
| Ours | 31.97 | 0.939 | 0.085 | 24.70 | 0.886 | 0.106 |

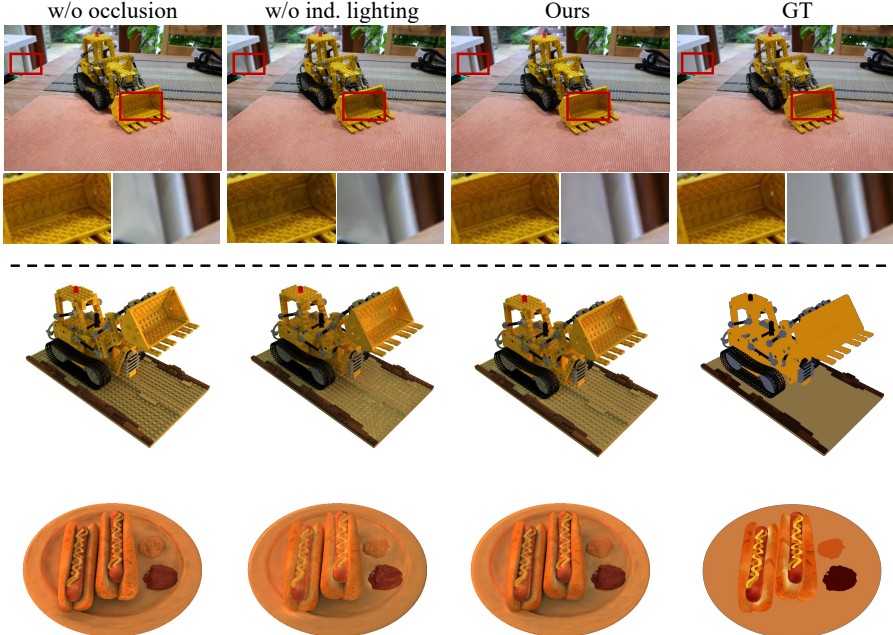

Figure 23: **Top**: Comparison of rendering results on the Mip-Nerf 360 dataset. **Bottom**: Comparison of albedo estimation on the TensoIR dataset.

## I.2 RESULTS ON THE STANFORD-ORB DATASET

To further evaluate the inverse rendering performance on real-world data, we conduct a comparison with GS-IR (Liang et al., 2024) on four scenes from the Stanford-ORB dataset. Table 11 presents the quantitative results. Our method outperforms GS-IR in novel-view synthesis, material estimation and relighting. Compared with GS-IR, our method estimates occlusion more accurately, thereby achieving a more effective decoupling of lighting and materials. Therefore, the estimated albedo contains fewer shadow and highlight components, resulting in better relighting quality. This is also supported by the qualitative comparisons in Fig. 27 and Fig. 28. We also visualize the relighting results under real-world environment map in Fig. 29.

## J ANALYSIS ON RELIGHTING

In this section, we analyze the impact of different settings on relighting results. It is well known that the accuracy of material estimation, especially the accuracy of albedo, is crucial to the quality of the relighting. Therefore, applying different transformations to the albedo will produce different relighting results. We find that some previous methods, such as TensoIR, will rescale the albedo using the ground truth in the dataset and normalize the energy of the environment map before relighting, then perform color gamut correction on the relighting results. Therefore, we conduct experiments on relighting under this setting and use Monte Carlo sampling to calculate the rendering equation. As illustrated in Fig. 30, compared to our original relighting results, using the settings in TensoIR can significantly improve the relighting performance.

Table 10: Quantitative results on the Synthetic4Relight dataset.

| | NVS | | | Albedo | | | Relighting | | | Roughness |
|---|---|---|---|---|---|---|---|---|---|---|
| | PSNR ↑ | SSIM ↑ | LPIPS ↓ | PSNR ↑ | SSIM ↑ | LPIPS ↓ | PSNR ↑ | SSIM ↑ | LPIPS ↓ | MSE ↓ |
| NeRFactor* | 22.80 | 0.917 | 0.151 | 22.96 | 0.906 | 0.162 | 21.54 | 0.875 | 0.171 | - |
| PhySG* | 23.42 | 0.987 | 0.068 | 21.80 | 0.973 | 0.185 | 22.63 | 0.973 | 0.073 | 0.268 |
| InvRender* | 26.19 | 0.991 | 0.044 | 25.25 | 0.983 | 0.058 | 25.59 | 0.984 | 0.041 | 0.072 |
| GS-IR | 34.79 | 0.973 | 0.042 | 23.20 | 0.918 | 0.091 | 26.30 | 0.945 | 0.071 | 0.214 |
| Ours | 35.42 | 0.973 | 0.042 | 24.68 | 0.931 | 0.085 | 27.36 | 0.945 | 0.070 | 0.128 |

Table 11: Quantitative results on the Stanford-ORB dataset.

| Scene | Method | NVS | | | Albedo | | | Relighting | | |
|---|---|---|---|---|---|---|---|---|---|---|
| | | PSNR ↑ | SSIM ↑ | LPIPS ↓ | PSNR ↑ | SSIM ↑ | LPIPS ↓ | PSNR ↑ | SSIM ↑ | LPIPS ↓ |
| Gnome | GS-IR | 37.73 | 0.984 | 0.023 | 22.57 | 0.886 | 0.101 | 24.83 | 0.905 | 0.069 |
| | Ours | 36.99 | 0.984 | 0.026 | 26.54 | 0.920 | 0.085 | 30.79 | 0.947 | 0.052 |
| Car | GS-IR | 36.61 | 0.990 | 0.010 | 26.21 | 0.958 | 0.033 | 24.91 | 0.959 | 0.031 |
| | Ours | 37.16 | 0.992 | 0.009 | 30.13 | 0.966 | 0.027 | 25.39 | 0.961 | 0.028 |
| Cactus | GS-IR | 35.91 | 0.987 | 0.016 | 32.81 | 0.962 | 0.038 | 30.42 | 0.974 | 0.018 |
| | Ours | 37.52 | 0.990 | 0.013 | 34.70 | 0.968 | 0.035 | 34.54 | 0.981 | 0.020 |
| Teapot | GS-IR | 35.10 | 0.987 | 0.013 | 28.46 | 0.956 | 0.037 | 21.07 | 0.962 | 0.024 |
| | Ours | 35.71 | 0.989 | 0.011 | 30.76 | 0.965 | 0.031 | 23.45 | 0.970 | 0.021 |
| Avg. | GS-IR | 36.34 | 0.987 | 0.015 | 27.51 | 0.940 | 0.052 | 25.31 | 0.950 | 0.036 |
| | Ours | 36.85 | 0.989 | 0.015 | 30.53 | 0.955 | 0.045 | 28.54 | 0.965 | 0.030 |

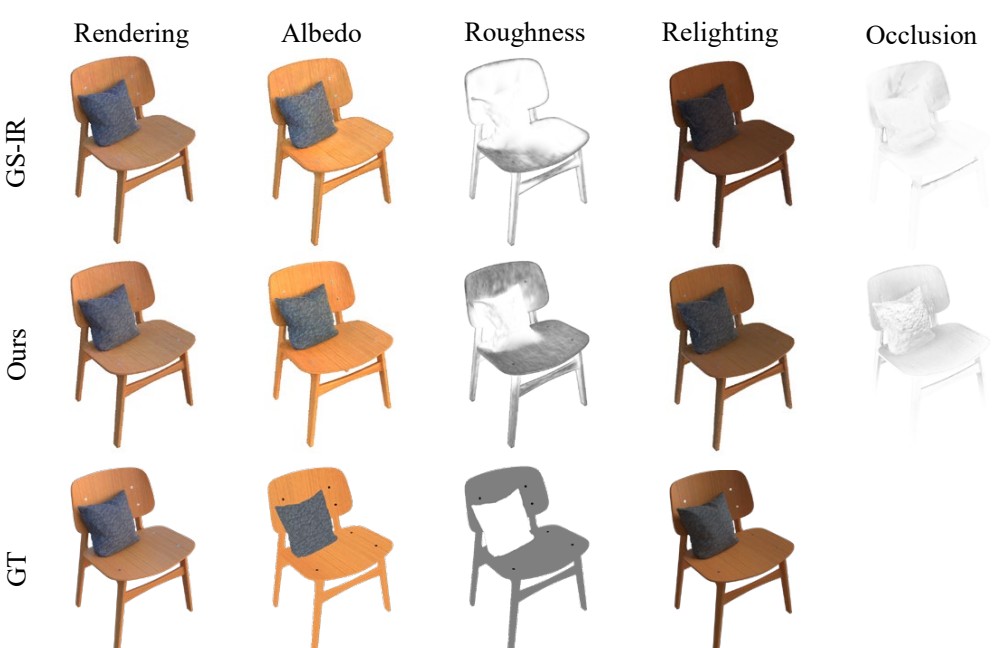

Figure 24: Qualitative comparison on Chair from the Synthetic4Relight dataset.

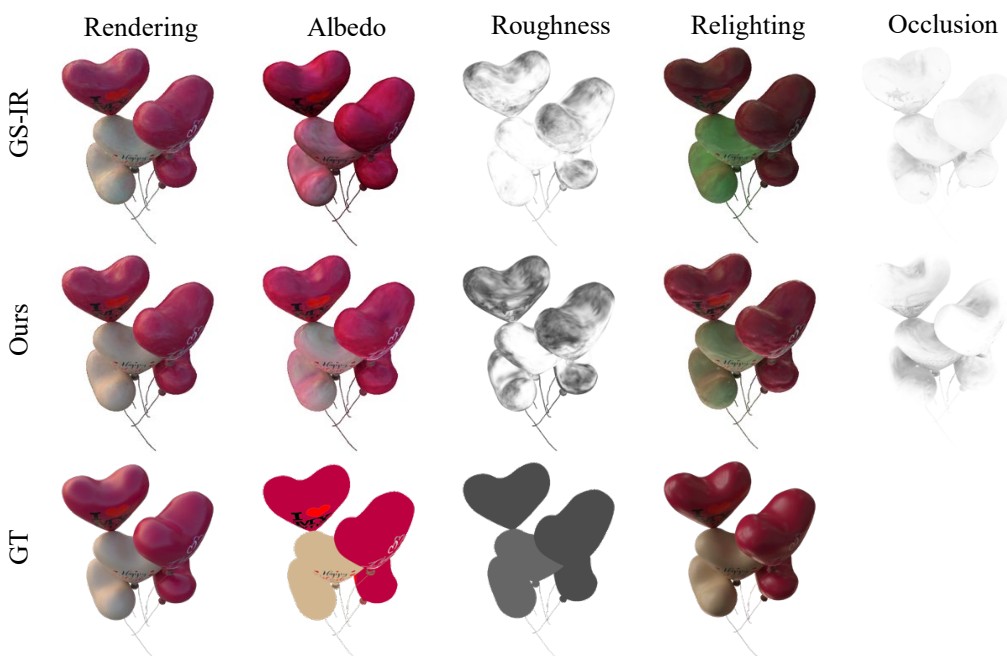

Figure 25: Qualitative comparison on Balloon from the Synthetic4Relight dataset.

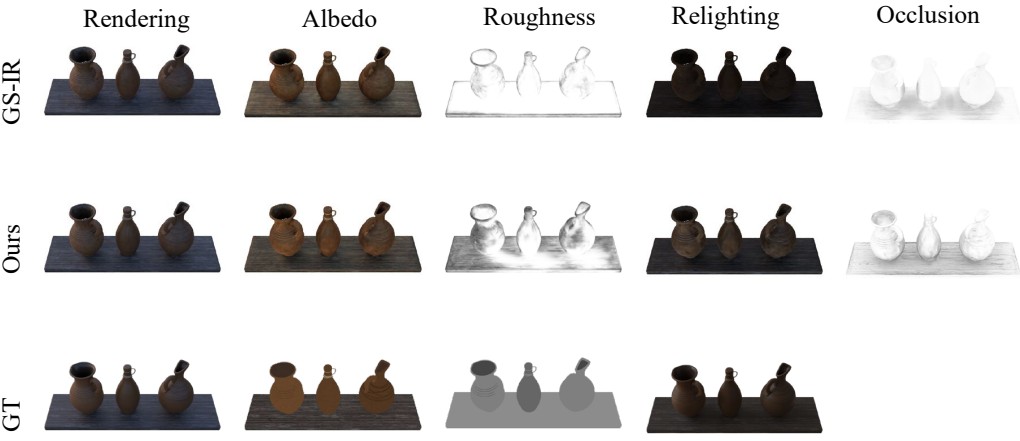

Figure 26: Qualitative comparison on Jugs from the Synthetic4Relight dataset.

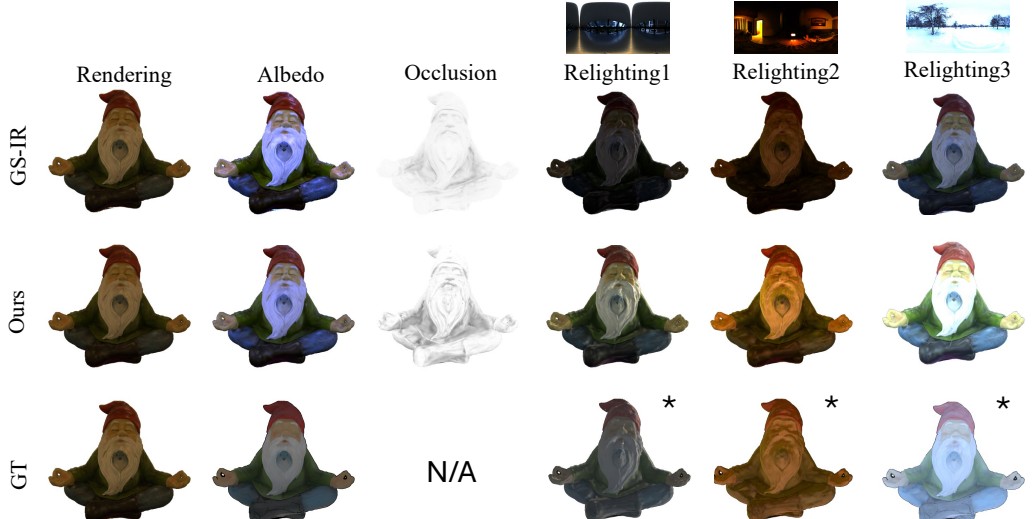

Figure 27: Qualitative comparison on Gnome from the Stanford-ORB dataset. The asterisk indicates that the GT relighting result is not available in the original dataset but is obtained by relighting using the albedo in the original dataset and the remaining material properties and normals estimated by the proposed method.

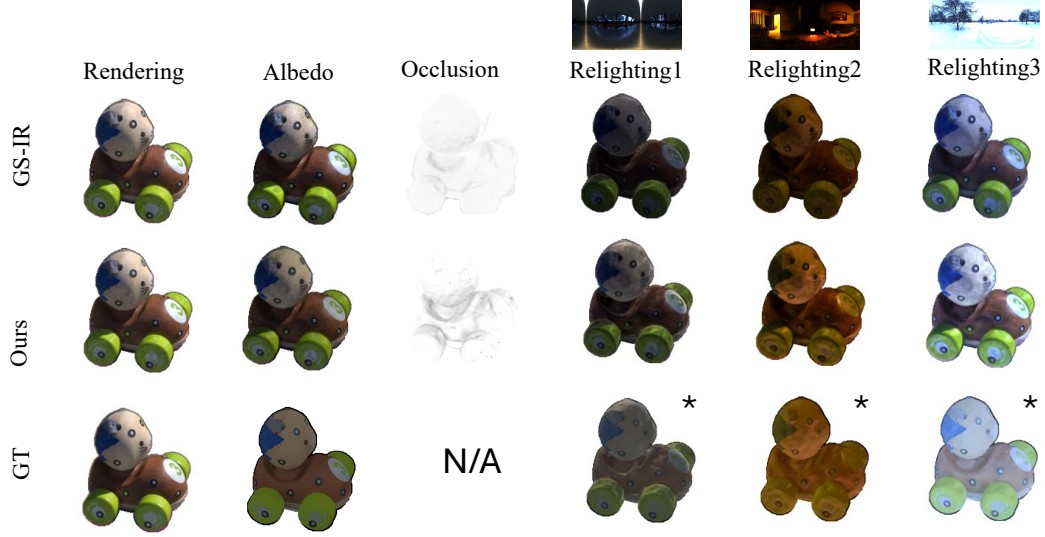

Figure 28: Qualitative comparison on Cactus from the Stanford-ORB dataset. The asterisk indicates that the GT relighting result is not available in the original dataset but is obtained by relighting using the albedo in the original dataset and the remaining material properties and normals estimated by the proposed method.

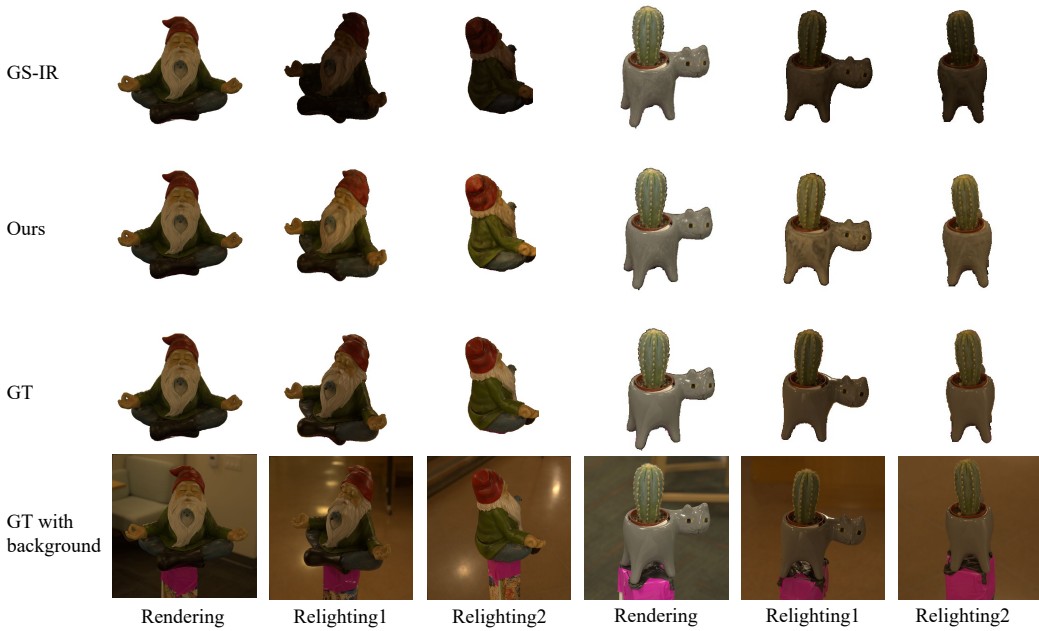

GS-IR

Ours

GT

GT with background

Rendering    Relighting1    Relighting2    Rendering    Relighting1    Relighting2

Figure 29: Qualitative comparison on relighting with real-world environment map.

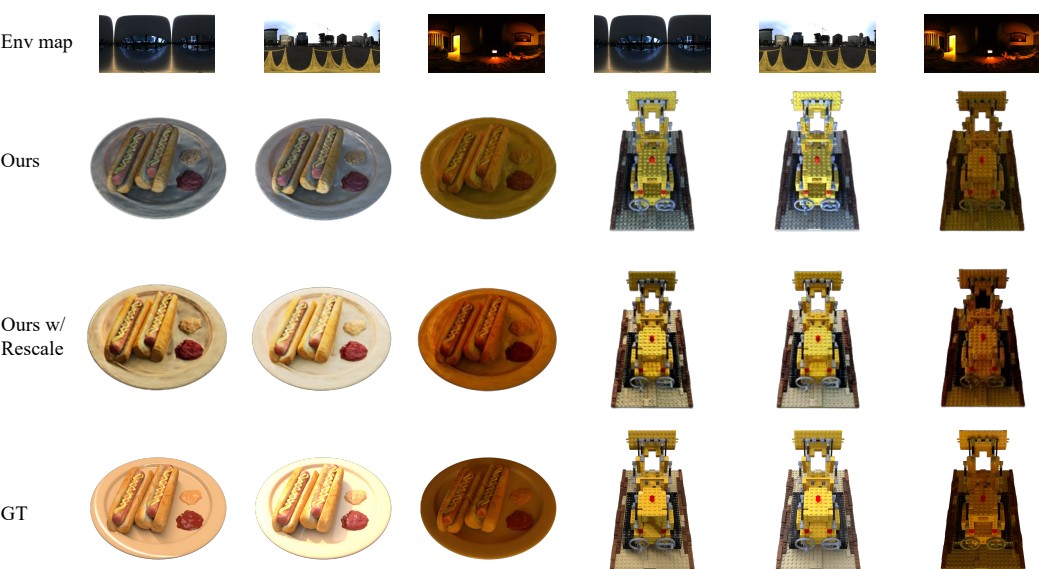

Env map

Ours

Ours w/ Rescale

GT

Figure 30: Qualitative comparison on relighting with different settings.

## K    GROUND TRUTH RELIGHTING RESULTS FROM THE TENSOIR DATASET

Fig. 31 provides the ground truth relighting results from the TensoIR dataset. The results demonstrate that the environment map used in Relighting 1 contains yellow light sources and dim blue ambient light, so the result of relighting is dominated by yellow tones. In contrast, when the environment map used is filled with bright white light sources and blue ambient light like in Relighting 2, the relighting result will have a cool tone.

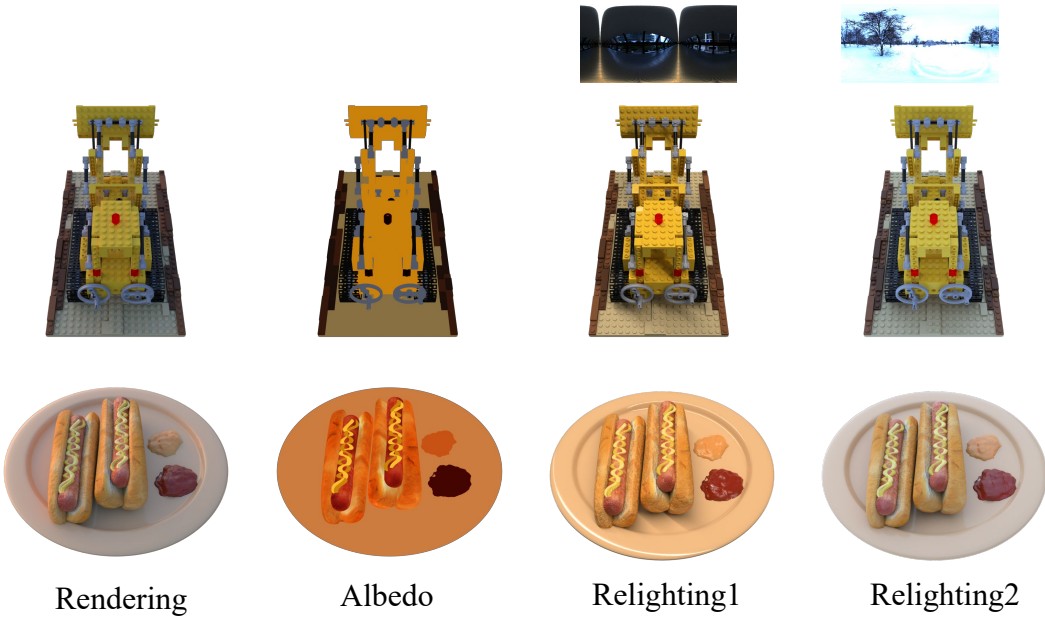

Rendering          Albedo          Relighting1          Relighting2

Figure 31: Ground truth relighting results from the TensoIR dataset.

