# OpenReview forum: "GI-GS: Global Illumination Decomposition on Gaussian Splatting for Inverse Rendering"
_ICLR.cc/2025/Conference — ICLR 2025 Poster_

### Official Review · Reviewer_uXjo · 2024-10-28

**Soundness:** 2
**Presentation:** 3
**Contribution:** 2
**Rating:** 6
**Confidence:** 5

**Summary:**

This work introduces a novel inverse rendering framework designed to achieve photo-realistic novel view synthesis and relighting by leveraging 3D Gaussian Splatting (3DGS) and deferred shading. Inverse rendering aims to accurately model scene materials, geometry, and lighting from images, a task complicated by the need to account for indirect lighting, which prior methods often model poorly. GI-GS improves upon these methods by incorporating efficient path tracing and deferred shading to compute indirect lighting, which leads to better handling of complex light-object interactions. The framework first uses a G-buffer to capture scene details and perform physically-based rendering (PBR) for direct lighting. Then, it applies lightweight path tracing to compute indirect lighting based on previous results, enabling more realistic global illumination. The authors demonstrate that GI-GS outperforms previous techniques in rendering quality and efficiency, making it a state-of-the-art method for novel view synthesis and relighting.

**Strengths:**

The paper is well-written and easy to follow, and it includes comprehensive relighting works from the past.

1. GI-GS uses path tracing with deferred shading to accurately model indirect lighting, achieving realistic relighting and view synthesis.

2. By combining 3D Gaussian Splatting with a physically-based pipeline, GI-GS efficiently reconstructs detailed geometry and materials for high-quality results.

3. The framework decouples direct and indirect illumination, enabling good-quality relighting across various conditions.

**Weaknesses:**

1. This method leverages image-based lighting (IBL) to simulate illumination from multiple directions, addressing both diffuse and specular reflections as outlined in Section 4.2. However, the decomposed diffuse and specular components are not evaluated, even in the supplementary material, which I believe is crucial for your lighting model.

2. The paper benchmarks against the TensoIR, Gaussian Shader and GS-IR, where these methods, including the relightable 3D Gaussian, offer accessible material evaluations. While the proposed method includes a material map (albedo, specular, roughness, etc.) in its pipeline, there should be an analysis or evaluation of the decomposed material, as material properties are central to reflection.

3. The paper claims that the calculated occlusion models shadowing, a key contribution similar to that of benchmarked methods. However, the experiments do not include a strong evaluation of this feature, despite visibility and occlusion information being available in the benchmarked methods.

4. Additionally, for indirect lighting, which accounts for bouncing and interreflection, there is no ground truth to verify the accuracy of the modeled indirect illumination. This could also be interpreted as the residual of your reflection during optimization.

I have to challenge the novelty and quality of the proposed method, as both occlusion and indirect lighting are also addressed and calculated similarly in these baseline methods and relighting evaluation didn't prove the proposed method achieved better quality than the previous method.

**Questions:**

1. How does the method validate the effectiveness of its decomposed diffuse and specular components, given that these are crucial for its lighting model but are not evaluated?

2. Given the importance of material properties for accurate reflection, why is there no analysis of the decomposed material map components (albedo, specular, roughness), especially since benchmarked methods include accessible material evaluations?

3. How does the paper substantiate its claim that calculated occlusion effectively models shadowing, considering that benchmarked methods provide visibility and occlusion data but no strong evaluation is presented for this feature?

4. Without ground truth for indirect lighting, how can the accuracy of modeled interreflection and bounce lighting be verified, and might these effects reflect residuals of reflection optimization?

5. In what ways does the proposed method’s relighting evaluation demonstrate superiority over existing methods, especially given that occlusion and indirect lighting calculations are also present in baseline methods?

---

> ### Author Response · Authors · 2024-11-24
> **Response to Reviewer uXjo - 1**
>
> We sincerely thank the reviewer for the detailed **assessment** of our submitted paper. The review comments are very insightful and have provided us with valuable guidance for further improving our work. We hope the following responses well address the concerns.
>
> ## 1. Evaluation of diffuse and specular components
>
> We appreciate the reviewer for raising this concern. There are two primary reasons why the decomposed diffuse and specular components were not evaluated in our study.
>
> First, there is **no ground truth** in existing datasets to quantitatively evaluate these components. Notably, widely used datasets such as TensoIR Synthetic, DTU, and Shiny Blender do not include ground truth data for diffuse and specular components. This limitation prevents us from performing quantitative evaluations of these specific components. In addition, prior inverse rendering studies, including the **seven baselines we compare against, similarly lack evaluations of these components.
>
> Second, conducting qualitative comparisons of the visualization results poses significant challenges. Our ability to assess rendering quality relies heavily on **real-world observations**. Generally, superior rendering results are those that closely approximate the appearance of real-world objects. However, diffuse and specular components cannot be decomposed in the real world, which means that we **lack understanding and prior knowledge** of them. Consequently, it is difficult to measure the quality of these results through human perception.
>
> In response to the reviewer’s request, we have included visualization results of the decomposed diffuse and specular components in **Fig. 21 in Appendix G** for further reference.
>
>
> ## 2. Material evaluation
>
> We thank the reviewer for this comment. In **Table 1** and **Fig. 4**, we provide both quantitative and qualitative comparisons of the recovered albedo on the TensoIR dataset. The results show that the albedo recovered by our method outperforms all baselines on the TensoIR dataset.
>
> It is important to note that the TensoIR dataset only provides ground truth of **albedo**, which limits our ability to assess other material properties such as specular and roughness. To address this limitation, we supplement the comparison results of material estimation on the Synthetic4Relight and Stanford-ORB datasets in **Appendix I**, which demonstrates the improvement of our method compared to baselines.
>
> ## 3. Occlusion evaluation
>
> We thank the reviewer for this comment. In **Fig. 6**, we present a comprehensive evaluation of the estimated occlusion using our method compared to GS-IR. The results demonstrate that our approach significantly outperforms the baseline at both **scene** and **object** levels. Specifically, our method captures fine details of occlusion, particularly in scene-level scenarios, where it effectively recovers the occlusion of distant objects. In addition, the insightful feedback from **Reviewer zyyD** further supports the effectiveness of our method. To strengthen our claim, we have included additional occlusion comparison results with further baseline methods in **Appendix F.1**.

---

> ### Author Response · Authors · 2024-11-24
> **Response to Reviewer uXjo - 2**
>
> ## 4. About indirect lighting
>
> We appreciate the insightful comment and the opportunity to clarify our perspective on this issue. Indirect lighting is a fundamental concept in computer graphics and vision, and it **should not** be regarded as merely a residual term in the lighting optimization process. Even with precise ground truth information of light sources and material properties, indirect light **persists** due to the inherent nature of light's multiple reflections within an environment.
>
> In some existing methods, including our baseline GShader, indirect lighting is treated as a residual term within the lighting optimization process. This simplification is primarily driven by the aim of reducing computational complexity, as accurately modeling multiple light reflections is highly challenging. However, this approach **limits** the representation of indirect lighting to the **specific lighting conditions present in the training data**. Consequently, it **fails** to generalize effectively during relighting. In contrast, our proposed method constructs indirect lighting based on the first-pass rendering results. This approach is equivalent to calculating indirect light from the outgoing radiance of the objects themselves, thereby enabling the computation of indirect lighting **under any given lighting conditions**. By doing so, our method does not rely on the residual term approximation and maintains the intrinsic characteristics of indirect illumination inherent to the environment.
>
> While the quality of indirect lighting cannot be directly measured due to the absence of ground truth data, we employ **indirect evaluation** methods to assess its effectiveness. Specifically, we evaluate the performance of material estimation and relighting tasks. Our method outperforms all baseline approaches in material estimation and surpasses other 3DGS-based methods in relighting, ranking second only to TensoIR. This performance indicates the effectiveness and accuracy of our indirect lighting model.
>
> It is important to note that our method's performance relative to TensoIR in relighting does not imply inferior quality in our material estimation or indirect lighting. The observed differences are primarily due to the **different rendering equation calculation processes** employed by each method. TensoIR achieves superior relighting results because it avoids the split-sum approximation for precomputing the environment map. Instead, it leverages **Monte Carlo integration** to accurately solve the rendering equation. For each surface point $x$, a set of rays is generated in various directions to **sample the environment map**, with visibility for each ray calculated separately using a visibility MLP. This approach allows TensoIR to compute the rendering equation with greater accuracy, leading to enhanced relighting results. However, this comes at the **cost of increased training and rendering times**.

---

> ### Author Response · Authors · 2024-11-24
> **Response to Reviewer uXjo - 3**
>
> ## 5. Concerns about the novelty and quality
>
> We thank the reviewer for raising this important question. Below, we explain the differences between our method and the baselines.
>
> The previous methods do take into account occlusion and indirect lighting, because global illumination is an important source of realism in the rendering results and is an important part of PBR. However, we claim that they obtain occlusion and indirect lighting in a very different way from our method.
>
> To obtain occlusion, we need to determine whether the ray intersects the surface. However, for volume rendering, this is very difficult due to the lack of an explicit definition of the surface. For NeRF-based methods, such as InvRender and TensoIR, they **use an MLP to estimate the visibility**. This paradigm is relatively simple to implement, but lacks clear geometric meanings. Among the 3DGS-based methods, GS-IR first renders the depth cube-map of a point in the 3D space from 3DGS, obtains the occlusion at that point from it, and then bakes the occlusion information into the grid for subsequent use. Although GS-IR also obtains occlusion from the depth, **it does not calculate the intersection of light and the surface**. Instead, it determines the occlusion relationship by comparing the depth value with a predetermined threshold, which can lead to inaccurate occlusion estimation. Moreover, the subsequent baking process uses spherical harmonics for further approximation, which introduces more errors. As for Relightable 3DGS (R3DG), it uses point-based ray tracing accelerated by the Bounding Volume Hierarchy (BVH) to obtain the occlusion and also bakes the occlusion in each Gaussian. However, as analyzed in our abstract, the distribution of 3D Gaussian primitives does not necessarily conform to the geometry of objects (especially at the scene level), **which limits the quality of point-based ray tracing and makes it difficult to extend to the scene level**.
>
> For indirect lighting calculation, NeRF-based methods often obtain it from the outgoing radiance fields parameterized by an MLP. As for 3DGS-based methods, such as GShader and R3DG, they model the indirect lighting **as a learnable attribute for each Gaussian**. Besides, GS-IR also uses a learnable  volume to store the indirect lighting.  However, these approaches do not model the multi-bounce of light and therefore **cannot construct indirect illumination during relighting**.
>
> Unlike previous methods, our method performs path tracing in the **screen space**, so it can more accurately determine the intersection of light and surfaces. It also reuses the information stored in the G-buffer, which reduces the amount of calculations. Moreover, our indirect illumination is obtained from the result of direct lighting, so it can be constructed during relighting. In addition, **by changing the path-tracing settings, our method can obtain global illumination with different effects** to adapt to different scenes and requirements, which is impossible for the baselines. (**See Fig.19 in Appendix F.4**)
>
>
>
> ## Q5. Relighting evaluation
>
> We thank the reviewer for this insightful question. As shown in Figure 4, our method demonstrates a closer alignment to the ground truth in terms of overall color tone for the **Lego** example compared to GS-IR. This improvement is largely due to our method's effective handling of indirect illumination, which enhances color accuracy and realism. For the **Hotdog** example, our method produces a notably smoother appearance with fewer artifacts, particularly in shadowed regions, such as the lower part of the plate and the middle sections of the hotdog and sauce.
>
> In addition, we have provided additional relighting evaluation on the Synthetic4Relight and Stanford-ORB datasets in **Appendix I**, which demonstrates the improvement of our method compared to baselines.

---

> ### Author Response · Authors · 2024-11-25
> **Official Comment by Authors**
>
> Dear Reviewer uXjo,
>
> We sincerely appreciate your time and effort in evaluating our work. We would greatly value the opportunity to engage in further discussion to see if our response solves the concerns. We have carefully addressed all of your insightful questions and hope that our responses help to better highlight the impact and results of our work. We kindly request you to review our responses and share any additional feedback or concerns, if any. We would be more than happy to address them further. Thank you!
>
> Best wishes,
>
> Authors

---

> ### Comment · Reviewer_uXjo · 2024-11-26
>
> Thank you for providing the detailed explanation regarding the evaluation of diffuse and specular components, material decomposition, and relighting.
>
> However, I believe there are issues in the separation of diffuse and specular components, as shown in Figure 21. Specifically, the specular reflection seems incorrectly modeled. **Specular reflection, primarily originating from light sources, should not exhibit significant color baked-in**. The results in Figure 21 suggest that your rendering equation fails to correctly represent specular reflection, resembling indirect illumination rather than specular reflection.
>
> Similarly, the material decomposition results in Figures 24 through 28 show limitations in evaluation. These figures focus primarily on diffuse albedo, which has already been validated in your main paper. The roughness evaluation does not sufficiently demonstrate the effectiveness of your method, and there is no specular evaluation presented in these figures **at all**. Additionally, the relighting results in Figures 27 and 28 are absent, making it difficult to properly assess the relighting capabilities of your method. Another question arises regarding Figure 27: the decomposed diffuse albedo contains strong blue tones, yet in Relighting 1, which features white light sources and a blue ambient, these blue tones are noticeably absent in the relighting results.
>
> As previously mentioned, my main concern lies in the unclear evaluation of material components, particularly diffuse and specular reflections, which are the primary reflections we perceive. Unfortunately, the supplementary results do not address this concern adequately.
>
> The issue with specular reflection is particularly critical because it represents high-frequency components, whereas diffuse reflection corresponds to low-frequency components. Existing baseline methods have already demonstrated successful inverse rendering for low-frequency recovery. If diffuse and specular separation is not conducted accurately, evaluating indirect illumination becomes valid only for predominantly diffuse scenes. However, your results include objects with strong specular components, such as the reflective bowls and onions on the desk in Figure 10, which further highlights the importance of correctly modeling specular reflection.

---

> > ### Author Response · Authors · 2024-11-27
> > **Response to Reviewer uXjo - 5**
> >
> > ## Response to Question 3
> >
> > As the reviewer noted, the reviewer's main concern lies in the unclear evaluation of material components, particularly diffuse and specular reflections. It is important to note that there may be some nuances in the terminology and concepts that require further clarification. In the context of inverse rendering, "material" typically refers to the intrinsic attributes of a surface, such as albedo, roughness, and metallic. In contrast, diffuse and specular reflections arise from the complex interplay between materials and lighting conditions, and thus cannot be directly equated with material properties. Therefore, the material estimation results presented in the main text and appendix (**See Tables 1, 8, 10, 11, Figure 4, and Figures 24-28**) do provide a detailed quantitative and qualitative evaluation of the estimated material properties. For the evaluation of diffuse and specular reflections, please refer to our detailed explanation in **Response to Question 2**.
> >
> > ## Response to Question 4
> >
> > We appreciate the reviewer’s insights but have a different perspective on this issue. The reviewer suggests that existing baseline methods have successfully demonstrated inverse rendering for low-frequency recovery. However, all previous representative inverse rendering methods mentioned in the **Response to Question 2** yield noticeable shadows in the estimated albedo. Importantly, for scenes or objects that are dominated by low-frequency components (such as diffuse reflection), **no current inverse rendering method has consistently outperformed the NVS method in both qualitative and quantitative comparisons**. In contrast, for scenes and objects with significant specular reflection, **methods that incorporate physically-based rendering, such as Ref-NeRF and Gaussian Shader, can outperform pure NVS methods in both quantitative and qualitative comparisons**. This indicates that accurately estimating low-frequency components can be more challenging than high-frequency components. This is because diffuse reflection necessitates consideration of incident light and visibility across the entire hemisphere, while specular reflection, particularly on shiny surfaces, can yield good results by accurately modeling just the light in the reflection direction. Moreover, it is important to note that most of the scenes and objects in our dataset have diffuse surfaces, which is representative of most real-world settings. Therefore, we believe that sacrificing some quality of specular reflection is an acceptable tradeoff. Nevertheless, we acknowledge the importance of accurately modeling specular components and will explore improvements in our future work.
> >
> >
> > Finally, we would like to use this opportunity to further clarify the novelty and contributions of our work. To the best of our knowledge, in the field of inverse rendering, there is currently no method that effectively balances efficiency and quality, works for both diffuse and specular surfaces, and extends to the scene level. Our approach focuses on improving efficiency and constructing global illumination for the diffuse component, and it can be applied at both the object and scene levels. Compared to previous methods, our approach demonstrates significant advantages in efficiency and is capable of constructing detailed occlusion and indirect illumination for both objects and scenes. However, we acknowledge some shortcomings in our method, such as the inaccurate estimation of the specular component and surface normals. We will refer to the feedback provided by all reviewers and make improvements in our future work.
> >
> > We genuinely appreciate the reviewer's insightful comments and assistance in improving our work. If our responses and the additional results sufficiently address your concerns, we kindly request your consideration for increasing your scores. Thanks once again for all the valuable feedback and understanding.

---

> > ### Author Response · Authors · 2024-12-01
> > **Request for feedback**
> >
> > Dear Reviewer uXjo,
> >
> > We sincerely appreciate the time and effort you have dedicated in the rebuttal period. We have carefully addressed all of your insightful questions and hope that our responses help to better highlight the impact and results of our work. We kindly invite you to review our responses and share any further feedback or concerns you may have. We would be more than happy to address them in more detail. If our responses and additional results have sufficiently addressed your concerns, we would greatly appreciate your consideration of raising your score. Thank you for your time and consideration！
> >
> > Best Regards,
> >
> > The Authors

---

> > > ### Comment · Reviewer_uXjo · 2024-12-02
> > >
> > > Thanks for the detailed response. I'll increase my score to board line accept

---

> > > > ### Author Response · Authors · 2024-12-02
> > > > **Thanks to the Reviewer**
> > > >
> > > > We deeply appreciate the reviewer’s kindness and we are also very thankful for the time spent by the reviewer during the rebuttal.

---

> ### Author Response · Authors · 2024-11-27
> **Response to Reviewer uXjo - 4**
>
> Thanks to the reviewer for the detailed review and the valuable insights the reviewer has provided.
>
> ## Response to Question 1
>
> For the incomplete separation of diffuse and specular components, we believe that the main reason is inadequate normal estimation.  Accurate normal estimation is very important for achieving high-quality specular reflections. Since our method does not use any geometric regularization, priors, or 3D representations with stronger geometric reconstruction capabilities like 2DGS, the estimated normals may not be very accurate. This inaccuracy can negatively affect the quality of the specular reflections observed in our results. To address this issue, we have provided a thorough analysis and a potential solution in our response to Reviewer **zyyD**. In addition, it is important to note that our method only considers diffuse indirect illumination, which may further affect the quality of specular reflection. However, this design choice primarily serves to improve efficiency, and we have included a detailed explanation in **Response to Reviewer SvCz - 3**.
>
> ## Response to Question 2
>
> We would like to clarify several points regarding the evaluation of our material decomposition.
>
> First, our roughness estimation results outperform those of GS-IR, which is the previous SOTA method based on 3DGS for inverse rendering. Furthermore, while our roughness estimation is second only to InvRender, it offers a significant advantage in **efficiency**, with a training speed that is **30 times faster** than InvRender.
>
> Regarding the evaluation of specular, if the reviewer is referring to metallic, we did not conduct an evaluation due to a lack of ground truth data in the dataset. If the concern relates to general specular reflection, we have explained the reason in our previous responses. Specifically, commonly used inverse rendering datasets such as TensoIR, Synthetic4Relight, ShinyBlender, GlossyBlender, and Stanford-ORB **do not provide ground truth for specular reflections**. Moreover, upon reviewing the representative inverse rendering research in recent years, including Neural Reflectance Fields, Nerv, Nerd, Nerfactor, Neilf, Physg, NVdiffrec, GS-IR, and R3DG, **we found that none of these studies have evaluated specular reflection**. This suggests that evaluating specular reflection remains a challenging task in the current state of the art. If the reviewer is aware of any prior work that has successfully evaluated specular reflection, we would appreciate the information. We would also be grateful for any guidance on methodologies for evaluating specular reflection.
>
> Regarding the relighting results shown in **Figures 27 and 28**, the absence of ground truth is due to the fact that the dataset does not provide relighting results for the specific environment map used. To address the reviewer's concern, we have included relighting results using the ground truth albedo provided in the dataset in our revised version. It is important to note, however, that the albedo provided by Stanford-ORB serves only as a pseudo-albedo representation of the actual object, which may affect the quality of the relighting results.
>
> Despite using the albedo from the dataset for relighting, blue tones do not appear in Relighting1. This is because the environment map corresponding to Relighting1 has a yellow light source in the lower part, while the ambient light, although blue, is relatively dim. As a result, the relighting output is dominated by yellow tones. In contrast, when using an environment map with a strong white light source and more prominent blue ambient light in Relighting3, the relighting results showcase clear blue tones. To avoid potential confusion, we have provided further analysis in **Fig. 31 in Appendix K**. According to the ground truth relighting results in the TensoIR dataset, it can be observed that for the environment map in Relighting1, the ground truth relighting results for both the Hotdog and Lego show an obvious yellow color, which further confirms our statement.

---

### Official Review · Reviewer_SvCz · 2024-10-29

**Soundness:** 2
**Presentation:** 3
**Contribution:** 2
**Rating:** 8
**Confidence:** 4

**Summary:**

This paper propose a efficient path tracing techinque that can calculate indirect lighting. They first render a G-buffer to capture the detailed geometry and material properties of the scene. Then, they perform physically-based rendering only for direct lighting. With the G-buffer and previous rendering results, the indirect lighting can be calcuated through the proposed path tracing. The training is decomposed into three stages: 1. vanilla 3dgs, 2.deferred rendering with split-sum approximation with occlusion calculated from the proposed path tracing, 3.add indirect lighting term on top of stage2.

**Strengths:**

The paper propose a novel path tracing strategy that can query visibility in screen space.

**Weaknesses:**

1. in line303-305, I think authors can refer to the uniform sampling in hemi-sphere used by NeILF, which should be better than  sampling in spherical coordinate
2. The authors argure that they are the frist to incorperate indrect illuminance in relighting, however, the proposed lighting model can not model the indirect lighting in specular term, also, the approximation in Equation7 makes it impossible to model sharp shadows as visbility is considered in the integration of direct lighting.
3. The path tracing part is not well introduced. I can't figure out how exactly is the procedure of sampling point on ray. I guess the authors mean they sample many points step by step until the depth of a point is smaller than the rendered depth.
4. The calculation of visibility by comparing the depth of sampled point on the ray with the rendered depth is not comprehensive. There is another condition: the ray (start from the surface) is not occluded by the object, but the sampled points on the ray has a bigger depth value than the rendered depth map. For example, a ray that has a consistent direction as the orientation of the camera, then whether it is occluded is not determined by the rendered depth.
5. I'm also confused about the calculation of indirect light, in line338-341. Is the outgoing radiation indicates $L_p(\hat{x},-\omega_)$? If so, why can it be obtained by indexing the RGB map. And in some conditon, indirect light of a surface point can not be seen by the camera. Equation13 is also confusing.

**Questions:**

see weakness.

---

> ### Author Response · Authors · 2024-11-24
> **Response to Reviewer SvCz - 1**
>
> We sincerely thank the reviewer for the detailed **assessment** of our submitted paper. The review comments are very insightful and have provided us with valuable guidance for further improving our work. We hope the following responses well address the concerns.
>
> ## 1.Sampling Strategy
>
> We sincerely thank the reviewer for this insightful and constructive suggestion. As utilized by NeILF, Fibonacci sampling has good properties, and we will incorporate this method in our future improvements. We have provided a discussion of this method in the revised paper and cited NeILF as a reference. (**See Appendix B, Lines 767-769**)
>
> ## 2. Specular term and sharp shadows
>
> We thank the reviewer for raising these critical points. Below, we provide detailed explanations to clarify our approach.
>
> **Why don't we consider specular terms?** The main reason is **efficiency**. Different from the diffuse component, accurately modeling the specular component of indirect lighting typically requires Monte Carlo sampling based on the GGX distribution. This process significantly increases computational complexity and reduces rendering speed. Moreover, in most cases, the contribution of the specular component to indirect lighting is minimal and often imperceptible. Therefore, omitting the specular component is a reasonable trade-off that maintains efficiency without substantially compromising visual fidelity.
>
> We acknowledge that the use of split-sum approximation in Eq. (7) leads to a certain limitation in modeling sharp shadows. This approximation necessitates pre-integrating the environment map, which prevents **per-ray sampling** of the original environment map. As a result, visibility cannot be accurately computed for each individual ray direction, thereby preventing the accurate rendering of sharp shadows. One potential solution is to abandon the split-sum approximation and use **Monte Carlo sampling** to calculate the rendering equation. This would allow us to calculate visibility for each sampled ray, enabling the modeling of sharp shadows. However, this approach would also lead to a significant increase in computational demands.
>
> Alternatively, our method remains effective for relighting scenarios under natural light sources, such as point light sources or area light sources. In these cases, we can employ **shadow map**, a widely used technique in computer graphics, to achieve precise shadows. Specifically, we can first generate a depth map from the light source's perspective. By comparing the depth of each shaded point with the corresponding value in the depth map, we can effectively ascertain visibility. We illustrate the rendering results under point light sources in **Fig. 20 in Appendix F.5**, where the shadows cast by the shapes are clearly visible.

---

> ### Author Response · Authors · 2024-11-24
> **Response to Reviewer SvCz - 2**
>
> ## 3&4. Path tracing and visibility calculation
>
> We appreciate the reviewer for highlighting the need for a clearer explanation of our path-tracing procedure. In response, we have revised **Equation 12** to enhance clarity regarding the visibility determination process, which may have previously led to some confusion. The reviewer's guess is basically consistent with our approach. However, we would like to clarify that the conditions for terminating the path-tracing process are somewhat **different**. Specifically, for each sampling point, **we not only compare its depth $z$ with the rendered depth $z_d$, but also compare its depth with $z_d+\delta$**, where $\delta$ represents the thickness of the surface and can be adjusted according to the scene. A ray is determined to have intersected the surface **if and only if** the sampled point satisfies $z_d < z < z_d+\delta$. This approach ensures a more precise detection of surface intersections by accounting for variations in surface thickness, thereby improving the robustness and accuracy of the path-tracing process.
>
> We hope this clarification addresses the reviewer’s concerns and enhances the understanding of our path-tracing methodology.
>
> Regarding the fourth question raised by the reviewer, we believe that the path-tracing strategy just explained can also address this concern. We have provided a detailed explanation in the **Appendix F.3, Lines 1140-1161**.
>
> ## 5. Indirect lighting calculation
>
> We appreciate the reviewer for raising this important question, which is similar to the third comment from **Reviewer tAkv.** Below, we provide detailed explanations to clarify our approach.
>
> In our first-pass rendering, the RGB values prior to tone mapping can be decomposed into diffuse and specular components as $I(\hat{u},\hat{v}) = L_o(\hat{x}, \omega_1) = L_d(\hat{x}) + L_s(\hat{x}, \omega_1)$, where the diffuse component $L_d(\hat{x})$ is isotropic and thus lacks directionality. The difference between $I(\hat{u},\hat{v})$ and the outgoing radiance $L_o(\hat{x}, -\omega)$ from $\hat{x}$ to $x$ is given by the specular component difference $\Delta L_s = L_s(\hat{x}, -\omega) - L_s(\hat{x}, \omega_1)$. In most practical scenarios, the specular component $\Delta L_s$ is relatively negligible compared to the diffuse component. Therefore, we believe that substituting $I(\hat{u}, \hat{v})$ to replace $L_o(\hat{x}, -\omega)$ serves as a practical approximation that avoids the need for additional calculations (**See Appendix F.2 for more details**). In addition, we have attempted to use only the diffuse component to calculate indirect lighting, but the experimental results are underperformed compared to our proposed strategy presented in the paper.
>
> Since our method is based on screen space, it cannot take into account indirect lighting outside the camera frustum. However, in most cases, the information available in screen space is **sufficient** for accurately constructing global illumination. Moreover, our method demonstrates a significant advantage in rendering speed compared with existing approaches. It is worth noting that most current game engines leverage similar techniques, utilizing information stored in the G-buffer to construct global illumination, which also aligns with our methodology.
>
> Moreover, we sincerely apologize for the confusion caused.  We have modified the notations in Eq. (13) and the main text to better distinguish the formulas related to $x$ and $\hat{x}$.

---

> > ### Comment · Reviewer_SvCz · 2024-11-25
> >
> > I appreciate the authors for providing such a detailed explanation in response to my questions. While the authors' response have already address most of my concerns, I still have some questions:
> > 1. GI-GS has already used monte carlo sampling for computing the visibility and indirect radiance in diffuse term, then why not applying the rendering equation without approximation for the diffuse term. I think the approximation in Eq.7 is not nessesary as GI-GS has already computed all the part needed for the integral.
> > 2. I can't find the defination of $N_s$ in line 520.

---

> > > ### Author Response · Authors · 2024-11-25
> > > **Response to Reviewer SvCz - 3**
> > >
> > > Thanks for your thoughtful feedback!
> > >
> > > We clarify that uniform sampling is used when calculating occlusion and indirect lighting. At the same time, we understand the approach you mentioned in the first question, that is, calculating both visibility and incident radiance for each sampled ray. However, the main problem lies in the **efficiency** of calculation. In our method, only **64 rays** are needed to calculate occlusion and indirect illumination. However, when using Monte Carlo sampling to calculate the rendering equation, in order to obtain a result with almost no noise, much more samples are required. Take TensoIR as an example. They sample every pixel in the environment map. Taking the **256*128** resolution they use as an example, the number of sampling rays required is **32768**, which is a much larger cost. Therefore, we use the split-sum approximation to calculate the rendering equation, and only use sampling to construct occlusion and indirect illumination.
> > >
> > > For the second question, $N_s$ is the number of sampled rays. And we used 64 in the experiment.
> > >
> > > Thanks again for your valuable review, which has been helpful in guiding our improvements.

---

> > > > ### Comment · Reviewer_SvCz · 2024-11-30
> > > >
> > > > Thanks for the detailed response. All of my concerns have been addressed, and I will raise my rating to "accept."

---

> > > > > ### Author Response · Authors · 2024-11-30
> > > > > **Thanks to the reviewer**
> > > > >
> > > > > We deeply appreciate the reviewer’s kindness and we are also very thankful for the time spent by the reviewer during the rebuttal.

---

### Official Review · Reviewer_tAKv · 2024-11-03

**Soundness:** 2
**Presentation:** 3
**Contribution:** 2
**Rating:** 6
**Confidence:** 4

**Summary:**

This works introduces GI-GS, a novel gaussian-splatting based inverse rendering pipeline to extract intrinsics from multi-view images. As a follow-up of existing GS-based inverse rendering,  GS-IR, this paper focuses on taking the indirect illumination into account, to solve more accurate material and environment map from photorealistic images.

**Strengths:**

The paper is intuitive and easy to follow.

Integrating indirect illumination calculation in a GS-based inverse rendering pipeline is a challenging movement.

In the experiments, the model shows comparable results in the task of novel view synthesis, decent performance in relighting, outperforming the GS-based baseline.

**Weaknesses:**

The way of processing indirect illumination in the pipeline is similar to InvRender: It uses the direct illumination (represented as a radiance function) of 3D points to calculate the indirect illumination. How is the indirect illumination calculation in this paper distincted from the one in InvRender?

Tab. 1: TensoIR performs much better than the proposed method in relighting task. and in figure 4 it is hard to see the differences between GS-IR and proposed method. This harms the soundness of the method. Please add more explanation of the performance.

The usage of I_{dir}(u, v) in the eq. 13 might be inaccurate, since the image RGB value is the combination of diffuse and specular lighting, which is not equal to the illumination from x_hat to x. One way to avoid this to re-apply the first rendering pass with new look-at direction. Is there any specific reason for directly using the RGB values? Please clarify.

While the main focus of the work is on decomposition and relighting, no relighting evaluation for real-world inputs is shown. This challenges the effectiveness of the method in more realistic and complex cases. An experiment conducted on real-world benchmarks (such as Stanford-ORB) can be very helpful to support the claims of the paper.

Other minor issues.
Adaptive Path Tracing happens in the projected camera only, which is not able to take the occluded objects into account.
Eq. (8): $\mathcal_{L}_{n-p}$ looks confusing (might be interpreted as n minus p), $\mathcal_{L}_{n,p}$ might be better.
Tab. 1: Only relighting methods are compared in the NVS column. Please consider adding reconstruction methods as reference.

Based on the above points, I’m on the fence but slightly leaning to reject this paper. I’ll carefully look at the authors’ responses and happy to change my score if the concerns are addressed.

**Questions:**

See Weaknesses.

---

> ### Author Response · Authors · 2024-11-24
> **Response to Reviewer tAKv - 1**
>
> We sincerely thank the reviewer for the detailed **assessment** of our method. The review comments are very insightful and have provided us with valuable guidance for further improving our work. We hope the following responses well address the concerns.
>
> ## 1. Differences from InvRender
>
> The indirect incoming radiance $L_i(x,\omega)$ of a surface point $x$ in direction $\omega$ is from the outgoing radiance $L_o(\hat{x},-\omega)$ at the ray intersection point $\hat{x}$ in the opposite direction $-\omega$.
>
> In **InvRender**, the outgoing radiance $L_o$ is modeled using an MLP. This approach inherently ties the outgoing radiance to the lighting conditions present in the training dataset, resulting in a fixed representation of outgoing radiance. Consequently, InvRender is limited in its ability to accurately reconstruct indirect illumination for novel lighting scenarios during relighting tasks, as the MLP cannot adapt the outgoing radiance beyond the training illumination conditions.
>
> In contrast, our method derives the outgoing radiance from the rendering result of the first pass, rather than relying on a fixed MLP model. This allows the outgoing radiance to dynamically reflect lighting conditions beyond those seen during training, facilitating more accurate and adaptable indirect illumination calculations during relighting. In addition, compared with the MLP-based method, our approach achieves superior computational efficiency and reduced storage requirements.
>
>
> ## 2. Relighting Evaluation
>
> We thank the reviewer for raising this concern. Below, we provide a detailed analysis of the relighting performance of TensoIR, GS-IR, and our proposed method.
>
> **Why TensoIR achieves high-quality relighting results**
>
> TensoIR achieves superior performance in relighting because it does not use the split-sum approximation typically used to precompute the environment map. Instead, TensoIR employs **Monte Carlo integration** to calculate the rendering equation during both the training and relighting phases. Specifically, for each surface point $x$, TensoIR generates a set of rays in various directions based on a distribution that samples the environment map. Visibility is then determined for **each individual ray** using a visibility MLP. This approach is similarly applied to compute indirect lighting from outgoing radiance. In this way, TensoIR can more accurately calculate the rendering equation, resulting in superior relighting quality. However, this high precision comes at the **cost of increased training and rendering times**, making TensoIR more computationally intensive.
>
> **Why the proposed method shows similar relighting results to GS-IR**
>
> GS-IR relies on caching occlusion and indirect illumination data under specific training lighting conditions within volumetric data. Consequently, these cached components become **less effective during relighting** under new lighting scenarios. However, in scenes with a single object, indirect lighting **only originates from the object itself**, which **limits** its impact on the final rendering results. For instance, in the case of a ball, there is no indirect lighting coming from itself. Therefore, relying solely on direct lighting does not significantly degrade the final rendering quality of GS-IR.
>
> Despite the similarities with GS-IR in single-object scenes, our method offers additional advantages. As shown in Figure 4, our method demonstrates a closer alignment to the ground truth in terms of overall color tone for the **Lego** example compared to GS-IR. This improvement is largely due to our method's effective handling of indirect illumination, which enhances color accuracy and realism. For the **Hotdog** example, our method produces a notably smoother appearance with fewer artifacts, particularly in shadowed regions, such as the lower part of the plate and the middle sections of the hotdog and sauce. In addition, we have also provided comparison with GS-IR on the Stanford-ORB and Synthetic4Relight datasets **in Appendix I**, which demonstrates the improvement of our method compared to GS-IR.

---

> ### Author Response · Authors · 2024-11-24
> **Response to Reviewer tAKv - 2**
>
> ## 3. Indirect illumination calculation
>
> We appreciate the reviewer for raising this insightful point. Below, we provide a detailed clarification for the use of $I_{dir}(\hat{x}, \hat{v})$ in Eq. (13).
>
> In our first-pass rendering, the RGB values prior to tone mapping can be decomposed into diffuse and specular components as $I(\hat{u},\hat{v}) = L_o(\hat{x}, \omega_1) = L_d(\hat{x}) + L_s(\hat{x}, \omega_1)$, where the diffuse component $L_d(\hat{x})$ is isotropic and thus lacks directionality. The difference between $I(\hat{u},\hat{v})$ and the outgoing radiance $L_o(\hat{x}, -\omega)$ from $\hat{x}$ to $x$ is given by the specular component difference $\Delta L_s = L_s(\hat{x}, -\omega) - L_s(\hat{x}, \omega_1)$. In most practical scenarios, the specular component $\Delta L_s$ is relatively negligible compared to the diffuse component. Therefore, we believe that substituting $I(\hat{u}, \hat{v})$ to replace $L_o(\hat{x}, -\omega)$ serves as a practical approximation that avoids the need for additional calculations (**See Appendix F.2 for more details**). In addition, we have attempted to use only the diffuse component to calculate indirect lighting, but the experimental results are underperformed compared to our proposed strategy presented in the paper.
>
> Besides, re-applying the first rendering pass with a new look-at direction may significantly increase the computational load, as the normal directions for different surface points vary, necessitating separate rendering passes for each distinct view direction. Such a process would not only be computationally intensive but also impractical for real-time applications or complex scenes.
>
> An alternative approach is to use Truncated Signed Distance Fields (TSDF) to rapidly reconstruct a mesh from 3DGS and subsequently perform path tracing to yield more accurate indirect lighting. However, constructing an accurate mesh at the scene level poses considerable challenges and introduces additional computational overhead.
>
>
> ## 4. Real-world Benchmark
>
> We appreciate the reviewer for this important aspect. We acknowledge that demonstrating the effectiveness of our method in realistic and complex scenarios is crucial. To address this, we have conducted experiments using the Stanford-ORB dataset.As shown in **Table 11, Fig. 27, and Fig. 28  in Appendix I.2**, our method achieves better albedo estimation and relighting than GS-IR. Specifically, our method estimates the albedo with fewer shadows and highlights and achieves usable relighting results. However, GS-IR still bakes lighting into albedo and performs poorly in relighting.
>
> Due to the lack of ground truth for relighting in the Stanford-ORB dataset, we have also conducted experiments on the **Synthetic4Relight** dataset. As shown in **Table 10, Fig. 24, Fig. 25, and Fig. 26  in Appendix I.1**, our method is also better than GS-IR in material estimation and relighting. In addition, it can be observed that the roughness estimated by our method has improved significantly compared with GS-IR, which is mainly due to the better occlusion estimation.
>
>
>
> ## 5. Other minor issues
>
> 1. Indeed, our adaptive path tracing operates within **screen space**, which inherently restricts its consideration to areas within the camera's frustum. Consequently, it does not account for objects that are completely occluded. However, in most cases, the information available in screen space is **sufficient** for accurately constructing global illumination. Moreover, our method demonstrates a significant advantage in rendering speed compared with existing approaches. It is worth noting that most current game engines leverage similar techniques, utilizing information stored in the G-buffer to construct global illumination, which also aligns with our methodology.
> 2. We thank the reviewer for pointing out this concern. We have modified Eq. (8) in the main text to avoid potential confusion.
> 3. To provide a more comprehensive evaluation, we have included the results of 3DGS in **Table 8 in Appendix D**. It is important to note that 3DGS was originally proposed for NVS. Besides, in most cases, the use of PRB in inverse rendering methods will negatively affect the quality of NVS. Therefore, for fairness, we only consider the inverse rendering methods when quantitatively comparing the NVS performance and use the results of 3DGS as a reference.

---

> > ### Comment · Reviewer_tAKv · 2024-11-24
> >
> > I appreciate the authors' thorough response and additional experiments that address my previous concerns. Upon reviewing the authors' replies and other reviews, I now better understand the paper's primary contribution: adapting G-buffer-based indirect illumination from real-time rendering to 3D inverse rendering.
> > The G-buffer approach, while not analytically precise for computing indirect lighting, represents a practical compromise that prioritizes computational efficiency. As both the authors and reviewer zyyD note, traditional Monte Carlo sampling methods achieve greater accuracy and closer alignment with ground truth results. Given this trade-off between efficiency and accuracy, I recommend that the authors:
> >
> > 1. Revise their claims to better reflect this efficiency-accuracy balance
> > 2. Place greater emphasis on the method's computational advantages
> > 3. Include a comparative analysis of training times against existing methods
> >
> > One additional suggestion regarding the Stanford-ORB results: The dataset includes ground-truth images for relighting, where objects are captured under various lighting environments with corresponding environment maps. Despite these images being captured from different viewpoints than the training data, they could provide valuable validation for the method's relighting capabilities. I encourage the authors to consider incorporating the relighting data into their evaluation.
> > Overall, I believe this work makes a decent contribution to the field, particularly in terms of computational efficiency, and happy to raise my score if the abovementioned issues are addressed.

---

> > > ### Author Response · Authors · 2024-11-25
> > > **Response to Reviewer tAKv - 3**
> > >
> > > Thanks for your thoughtful feedback! We sincerely appreciate your evaluation and recognition of our contributions, which helped us identify certain inappropriate statements in the previous version of the paper. As a result, we have revised the contribution section and made adjustments to some statements in other parts of the paper. **Specifically, we have revised the following parts in the paper**:
> > >
> > > - Abstract, Lines 26 - 29.
> > > - Contribution of our method, Lines 92 - 102.
> > > - Add training time in Table 1, Lines 378 - 400.
> > > - Experiment results, Lines 414 - 415
> > > - Conclusion, Lines 533 - 534.
> > >
> > > We also provide the relighting result of Stanford-ORB dataset using real-world environment map in **Table 11 and Fig. 29 in Appendix I.2**, which shows the improvement of our method compared to GS-IR.
> > >
> > > In addtion, we have conducted further analysis on the relighting performance. After checking the official code of TensoIR, we found that in the relighting experiment, TensoIR rescales the predicted albedo using the ground truth albedo in the dataset and normalizes the energy of the environment map. In addition, it also performes gamma correction on the results of relighting. This series of operations helps to improve the performance of relighting, so we conducted additional experiments under the setting of TensoIR and used Monte Carlo sampling to calculate the rendering equation. The results show that the above settings can significantly improve the relighting performance. I hope this helps to further explain the issues you mentioned earlier about the quality of relighting results. (**See Appendix J**).
> > >
> > > Thanks again for your valuable suggestion, which has been helpful in guiding our improvements.

---

> > > ### Author Response · Authors · 2024-11-27
> > > **Response to Reviewer tAKv - 4**
> > >
> > > Dear Reviewer tAKv,
> > >
> > > We sincerely appreciate your time and effort in evaluating our work. We would greatly value the opportunity to engage in further discussion to see if our responses solve your concerns. We have carefully addressed all of your insightful questions and hope that our responses help to better highlight the impact and results of our work. If our responses and the additional results sufficiently address your concerns, we kindly request your consideration for increasing your scores. Thank you once again for all your valuable feedback and understanding.
> > >
> > > Best wishes,
> > >
> > > Authors

---

> > > > ### Comment · Reviewer_tAKv · 2024-12-01
> > > >
> > > > I thank the authors' effort in answer my questions and revising the paper. I have no more questions, and am happy to raise my score.

---

> > > > > ### Author Response · Authors · 2024-12-01
> > > > > **Thanks to the reviewer**
> > > > >
> > > > > We deeply appreciate the reviewer’s kindness and we are also very thankful for the time spent by the reviewer during the rebuttal.

---

### Official Review · Reviewer_zyyD · 2024-11-04

**Soundness:** 3
**Presentation:** 4
**Contribution:** 3
**Rating:** 8
**Confidence:** 4

**Summary:**

The paper present a novel inverse rendering framework, GI-GS, to achieve scene decomposition and photo-realistic relighting.

Existing 3DGS-based methods, due to their nature of non-deterministic geometry, encounter challenges in modeling inter-reflection and self-occlusion, which prevent them from occlusion-aware/indirect-light-aware decomposition or relighting. While recent works like relightable 3DGS use ray tracing to estimate occlusion, preventing shadow from being baked into materials, many of them are still unable to process inter-reflection, or only employ a learnable residual term to model indirect light.

To address this, the paper propose a novel framework with deferred rendering technique, which employs path tracing to achieve indirect illumination computation. Experimental results are provided to demonstrate the effectiveness of the framework, highlighting the promising potential of deferred rendering in efficiently modeling indirect geometry-lighting interactions. In object-level cases, the proposed method outperforms several baselines in both novel view synthesis and decomposition, while achieving comparable performance in relighting. In scene-level cases, the method also demonstrates meaningful results in both decomposition and relighting.

**Strengths:**

1. The paper propose a novel framework where the deferred rendering technique enables light-weight path tracing on the deferred rendering buffer, which highlights the promising potential of deferred rendering in efficiently modeling indirect geometry-lighting interactions.
2. The method is extended to scene level and successfully produces meaningful decomposition and relighting results.
3. In novel view synthesis tasks, the method achieves best performance over inverse rendering baselines.

**Weaknesses:**

1. The novel indirect illumination module may require further experimentation to validate its effectiveness. Theoretically, this module has the potential to achieve superior relighting performance compared to solutions that rely on learnable parameters for modeling indirect light, as the fixed inter-reflection in those methods is often inaccurate when relighted under another illumination.

However, the quantitative results presented in Table 1 indicate that the relighting performance is subpar, even when achieving best albedo estimation, which undermines the strength of the paper's key claim. I believe the relighting performance is likely impacted by the inadequate normal estimation, as illustrated in Fig. 9 (Garden, Kitchen, Counter, and Bonsai), rather than suggesting that path tracing is ineffective, especially since the impressive qualitative results in Fig. 6 indicate otherwise.

Therefore, to quantitatively validate the effectiveness of the method, it would be beneficial to conduct an ablation study focused on albedo estimation or relighting. The current ablation study in Table 3 only examines performance changes related to novel view synthesis quality, which is insufficient to support the claims made in the paper.

2. In the absence of quantitative results, the qualitative findings suggest that normal estimation may be inadequate, as demonstrated in Fig. 9 (Garden, Kitchen, Counter, and Bonsai). This deficiency could negatively impact the overall performance of inverse rendering.

**Questions:**

1. The occlusion map and indirect light map in Figs. 6 and 8 are quite impressive. However, I am curious as to why Fig. 4 shows that the proposed method still retains a significant amount of shadow in the albedo, particularly in the case of the Hotdog.

2. What does the direct light look like? The incorrect direct lighting reconstruction also lead to poor relighting performance.

---

> ### Author Response · Authors · 2024-11-24
> **Response to Reviewer zyyD - 1**
>
> We sincerely thank the reviewer for the detailed **assessment** and **recognition** of our method. The review comments are very insightful and have provided us with valuable guidance for further improving our work. We hope the following responses well address the concerns.
>
> ## 1. Ablation study on albedo and relighting
>
> We appreciate the reviewer's insightful suggestion and have conducted additional ablation studies on albedo estimation and relighting to demonstrate the effectiveness of occlusion and indirect illumination in our method.
>
> Our experiments show that incorporating occlusion and indirect illumination enhances the decoupling of lighting and materials. The quantitative results demonstrate that both occlusion and indirect illumination have a significant impact on albedo estimation and relighting quality. Specifically, the lack of occlusion leads to more pronounced shadows in the albedo, while overlooking indirect illumination results in noticeable over-bright areas in the albedo. (**See Table 9 and Fig. 23 in Appendix H**)
>
> ## 2. Inadequate normal estimation
>
> We sincerely thank the reviewer for the constructive comments regarding the normal estimation quality. Below, we provide the reasons for inadequate normal estimation and a discussion on how to improve its quality.
>
> It is important to note that our method does **not rely on additional geometric priors or regularization**. As a result, the reconstructed normals are not perfectly accurate, which subsequently affects the overall performance of our method. In addition, our 3DGS-based baseline GS-IR also does not incorporate additional regularization terms or priors to supervise the normals. Therefore, the experimental results under the same setting ensure a **fair comparison** of occlusion and indirect illumination qualities between the methods.
>
> To demonstrate that the inadequate normal estimation negatively affects the relighting results, we use a **pre-trained Omnidata model** for normal supervision. Experimental results show that incorporating additional priors to obtain more accurate normals significantly enhances the relighting quality of our method.  (**See Figures 14 and 15 in Appendix E, Lines 1029-1079**)
>
> In order to improve the quality of normal estimation, a straightforward approach is to use the shortest axis of the 3D Gaussian as the normal direction and apply additional regularization to make the Gaussian shape close to a disk. However, we observe that the introduction of regularization limits the representation capability of the 3DGS and results in the loss of details in complex geometric areas when reconstructing real-world scenes. Notably, some recent works [1] [2] have pointed out that **optimizing BRDF and normal together** can significantly improve the normal quality without sacrificing the representation capability of 3DGS. We believe that this paradigm offers a more effective means of supervising normals, and we plan to incorporate it in our future work.
>
> [1] Wei M, Wu Q, Zheng J, et al. Normal-GS: 3D Gaussian Splatting with Normal-Involved Rendering[J]. arXiv preprint arXiv:2410.20593, 2024.
>
> [2] Liang Z, Li H, Jia K, et al. GUS-IR: Gaussian Splatting with Unified Shading for Inverse Rendering[J]. arXiv preprint arXiv:2411.07478, 2024.

---

> ### Author Response · Authors · 2024-11-24
> **Response to Reviewer zyyD - 2**
>
> ## Q.1 Shadow in the albedo
>
> We thank the reviewer for raising this important question. Below, we provide a detailed explanation of the presence of shadows in albedo, as well as potential solutions.
>
> The shadows observed in the albedo come from the way we construct global illumination using **screen space** information and the **use of split-sum approximation**. Since the information is derived from the camera’s perspective, any indirect light or occlusion originating from areas outside the camera frustum may not be accurately represented. This limitation contrasts with the path-tracing method applied directly on the mesh, which typically yields more precise results. We choose this screen space approach primarily to enhance computational efficiency and to accommodate scenarios where mesh data may be unavailable or incomplete.
>
> Furthermore, since we use split-sum approximation to calculate render equation, which assumes that the intensity of incident radiance in different directions is consistent. Under this assumption, occlusion obtained by integrating visibility represents the degree to which the point is occluded. Consider **an example** where there is a point light source in the center of an empty room. In this case, the points at the corners are not occluded in the direction of the light source, but the occlusion obtained by integration is 0.75. This shows that when the light source is anisotropic or has significant directionality, this estimation is often inaccurate. Using **Monte Carlo sampling** for the sampling ray in each direction and calculating the visibility separately can obtain a more accurate shadow, but it will significantly sacrifice the computational efficiency.
>
> ## Q.2 Direct lighting visualization
>
> We fully agree with the reviewer's point of view that direct lighting has a significant impact on rendering results. To illustrate this, we have visualized the direct lighting of indoor scenes from the Mip-NeRF dataset during relighting under conditions with and without normal supervision (**See Figure 22 in Appendix G, Lines 1258-1277**). It can be seen that incorrect direct lighting can severely compromise the quality of relighting results. This degradation is mainly caused by inaccurate normal estimation.

---

> > ### Comment · Reviewer_zyyD · 2024-11-24
> >
> > I greatly appreciate the authors' clarification and the comprehensive explanation provided in the appendix, which significantly enhances the community's understanding of screen-space path-tracing techniques in the context of 3DGS-grounded inverse rendering. With my concerns (W1, W2) adequately addressed, I believe this novel work makes a meaningful contribution. However, there remains a minor issue that impacts the overall soundness of the paper, which I would like to discuss below.
> >
> > ---
> >
> > The paper claims state-of-the-art decomposition performance in L097. However, I believe this claim is not sufficiently supported by either the quantitative or qualitative results presented:
> >
> > 1. In Table 1 and Figure 4, despite the higher albedo PSNR (which is not convincing enough to demonstrate decomposition ability as the albedo scaling can be quite strong), the decomposition of GI-GS appears flawed, leading to significant degradation in relighting performance, particularly when compared to TensoIR.
> >
> > 2. In Figures 24, 25, and 26, some of the albedo, roughness, and relighting outputs are noticeably incorrect. While the results are an improvement over GS-IR, they fall short of convincingly demonstrating that GI-GS achieves SOTA decomposition performance, especially given the absence of other decomposition baselines, such as TensoIR and Relightable 3DGS.
> >
> > In light of these observations, I respectfully suggest the authors revise their claims to more precisely reflect the paper's contributions, particularly in areas where the experimental results are insufficient to provide robust evidence. Furthermore, it would be beneficial to emphasize the impressive strengths of GI-GS, such as its non-object-level decomposition capability (compared to methods that seem unsuitable for unbounded scenes, such as TensoIR) and its remarkable efficiency. Addressing these points would enhance the paper's soundness, and I would raise my rating in that case.

---

> > > ### Author Response · Authors · 2024-11-25
> > > **Response to Reviewer zyyD - 3**
> > >
> > > Thanks for your thoughtful feedback! We are very grateful for your evaluation and recognition of our contribution, which helped us realize some inappropriate statements in the previous version of the paper. Therefore, we have revised the contribution section in the paper, as well as some statements in other parts. **Specifically, we have revised the following parts in the paper**:
> > >
> > > - Abstract, Lines 26 - 29.
> > > - Contribution of our method, Lines 92 - 102.
> > > - Add training time in Table 1, Lines 378 - 400.
> > > - Experiment results, Lines 414 - 415
> > > - Conclusion, Lines 533 - 534.
> > >
> > > In addition, we have conducted further analysis on the relighting performance. After checking the official code of TensoIR, we found that in the relighting experiment, TensoIR rescales the predicted albedo using the ground truth albedo in the dataset and normalizes the energy of the environment map. In addition, it also performes gamma correction on the results of relighting. This series of operations helps to improve the performance of relighting, so we conducted additional experiments under the setting of TensoIR and used Monte Carlo sampling to calculate the rendering equation. The results show that the above settings can significantly improve the relighting performance. I hope this helps to further explain the issues you mentioned earlier about the quality of relighting results. (**See Appendix J**). We also provide the relighting results of Stanford-ORB dataset using real-world environment map. (**See Table 11 and Fig. 29 in Appendix I.2**).
> > >
> > > Thanks again for your valuable suggestion, which has been helpful in guiding our improvements.

---

> > > > ### Comment · Reviewer_zyyD · 2024-11-25
> > > >
> > > > I appreciate the authors' efforts in providing such inspiring work, and I believe it significantly enhances the community's understanding of screen-space path-tracing techniques in the context of 3DGS-grounded inverse rendering, although the method may not achieve the best inverse rendering performance for all cases. Accordingly, I am happy to raise my rating.

---

> > > > > ### Author Response · Authors · 2024-11-25
> > > > > **Thanks to the reviewer**
> > > > >
> > > > > We greatly appreciate the kindness of the reviewer, and we are also very thankful for the time spent by the reviewer during the rebuttal.

---

### Author Response · Authors · 2024-11-24
**General Response**

# Dear Reviewers, ACs, and PCs,

Thank you very much for your dedication, support, and insightful feedback. We are delighted with your recognition of the quality of our global illumination [zyyD] and your suggestions for improvement [SvCz]. We sincerely appreciate all your insightful comments [zyyD, tAKv, uXjo, SvCz]. We have reviewed all the comments, addressed all questions, and provided addtional experimental results. Below, we summarize the revisions we made:

## Revisions and Updates

### Additional Experimental Results

- [zyyD] Ablation study on material estimation and relighting. (**Appendix H**)
- [zyyD] Analysis of the relationship between normal estimation and relighting. (**Appendix E**)
- [zyyD] Visualization of direct lighting during relighting. (**Fig. 22 in Appendix G**)
- [tAkv] Adding 3DGS as a reference (**Table 8 in Appendix D**)
- [tAKv, uXjo] Additional comparisons on the Stanford-ORB and Synthetic4Relight datasets. (**Appendix I**)
- [SvCz] Sharp shadows under natural light source. (**Appendix F.5**)
- [uXjo] Additional occlusion comparison with TensoIR and GS-IR. (**Appendix F.1**)
- [uXjo] Decomposed diffuse and specular components. (**Fig. 21 in Appendix G**)
- Controllability of global illumination (**Appendix F.4**)

### Writing Standards and Clarity

- [tAKv, SvCz] Analysis of indirect lighting calculation. (**Appendix F.2**)
- [tAKv] Changed subscript in **Eq. (8).**
- [SvCz] Corrected **Eq. (12)** and added explanation in **Appendix F.3**.
- [SvCz] Modified the notations in **Eq. (13)**.

### Discussion and Citation

- [SvCz] Added the discussion about Fibonacci sampling used in NeILF. (**Appendix B**)

## Request for Feedback

We respectfully invite the reviewers to carefully evaluate our revisions and the individual responses provided. We are more than willing to address any remaining questions or concerns. If our responses and the additional results sufficiently address your feedback, we kindly request your consideration for increasing your scores. We sincerely appreciate your thoughtful engagement and constructive suggestions, which have been instrumental in enhancing the quality of this work.

**Best Regards,**

_The Authors_

---

> ### Author Response · Authors · 2024-11-25
> **New revisions to the paper**
>
> We are very grateful to the reviewers **zyyD** and **tAKv** for their comments and suggestions. As a result, we have revised some of the contents of the paper. **Specifically, we have revised the following parts in the paper**:
>
> - Abstract, Lines 26 - 29.
> - Contribution of our method, Lines 92 - 102.
> - Add training time in Table 1, Lines 378 - 400.
> - Experiment results, Lines 414 - 415
> - Conclusion, Lines 533 - 534.
>
> Moreover, we have provided further analysis of the relighting performance in **Appendix J** and additional relighting results of Stanford-ORB dataset in **Table 11** and **Fig. 29**.
>
> Thanks again to all the reviewers for their comments, which contributed to the further improvement of our work.

---

### Meta-Review · Area_Chair_dtNF · 2024-12-20

**Metareview:**

This paper presents a new 3D Gaussian Splatting method that takes into account the indirect illumination to achieve photo-realistic novel view synthesis and relighting. The strength of the paper is the introduction of a new path tracing strategy that can query visibility in the screen space for indirect illumination handling. This merit is something reviewers consistently agreed on. On the other hand, there were concerns about the unclear evaluation, particularly diffuse and specular reflections. This stemmed from the lack of the ground truth for quantitative evaluation. This respect has been thoroughly discussed during the author-reviewer discussion phase, and the reviewers were convinced about the additional results. The AC agreed with the reviewers' positive opinions and rendered this recommendation.

**Additional Comments On Reviewer Discussion:**

The unclear experimental evaluation was the chief concern raised at the review phase. The issue has been resolved over extensive discussions between the reviewers and authors by the additional materials showing the qualitative and quantitative evaluation.

---

### Decision · Program_Chairs · 2025-01-22

Accept (Poster)